# Strongly anharmonic flux-tunable transmon based on InAs-Al 2D heterostructure

Shukai Liu[1,5] ✉, Arunav Bordoloi [1,2,5] ✉, Jacob Issokson[2], Ido Levy[2], Maxim G. Vavilov[3], Javad Shabani[2] & Vladimir E. Manucharyan [1,4]

The gatemon qubits, made of transparent superconducting-semiconducting Josephson junctions, typically have even weaker anharmonicity than the opaque AlOx-junction transmons. However, flux-frustrated gatemons can acquire a much stronger anharmonicity, originating from the interference of the higher-order harmonics of the supercurrent. Here we investigate this effect of enhanced anharmonicity in split-junction gatemon devices based on InAs-Al 2D heterostructure. We find that anharmonicity in excess of 100% can be routinely achieved at the half-integer flux sweet-spot without any need for electrical gating or excessive sensitivity to the offset charge noise. We verified that such intrinsically large anharmonicity enables our devices to be driven coherently with raw Rabi frequencies exceeding 100 MHz, without any pulse shaping, simplifying implementation and control compared to traditional gatemons and transmons. Furthermore, by analyzing a relatively high-resolution spectroscopy of the device transitions as a function of flux, we were able to extract fine details of the current-phase relation, to which transport measurements would hardly be sensitive. The strong anharmonicity of our anharmonic tunable transmons, along with their bare-bones design, can prove to be a precious resource that transparent superconducting-semiconducting junctions bring to quantum information processing.

Electrically-gated transmon qubits (gatemons) have been demonstrated as potentially useful devices for building large scale quantum processors[1–15]. A typical gatemon device is made with a gate electrode close to a hybrid superconductor-semiconductor Josephson junction (superconducting-semiconducting JJ). The voltage on the gate electrode changes the electron density in the semiconductor weak link and thereby tunes the Josephson energy $E_J$ of the qubit. The advantage of electrical gating is that it can be used for controlling the quantum dynamics by rapidly changing the qubit frequency, similarly to switching transistors in conventional semiconductor circuits. Just like transmons, however, gatemons suffer from a suppressed anharmonicity of the qubit transition, as they operate in a regime where the

Josephson phase-difference is localized near the bottom of the Josephson potential well[16–20]. Increasing the anharmonicity, which we define by

$$\eta = \left| \frac{f_{12} - f_{01}}{f_{01}} \right|, \tag{1}$$

would require increasing the charging energy $E_C$, which would exponentially increase the qubit's sensitivity to the offset-charge noise. Furthermore, the relatively high transparency of superconducting-semiconducting junctions further reduces anharmonicity in comparison to opaque AlOx transmon junctions[21]. The weak anharmonicity of

[1]Department of Physics, Joint Quantum Institute, and Quantum Materials Center, University of Maryland, College Park, MD, USA. [2]Center for Quantum Information Physics, Department of Physics, New York University, New York, NY, USA. [3]Department of Physics, University of Wisconsin-Madison, Madison, WI, USA. [4]Institute of Physics, Ecole Polytechnique Federale de Lausanne, Lausanne, Switzerland. [5]These authors contributed equally: Shukai Liu, Arunav Bordoloi. ✉e-mail: sliu499@umd.edu; ab13024@nyu.edu

gatemons would limit gatemon-based quantum processors in the same way it limits transmons, that is single-qubit and two-qubit gate operations will have to be done sufficiently slowly as to prevent leakage outside the computational space[22,23].

Replacing the gatemon junction with a split junction and applying a flux bias through the loop can dramatically increase the qubit anharmonicity in comparison to an equivalent split-junction transmon device[2,24]. This effect originates from a destructive interference of the first Josephson harmonics and a constructive interference of the typically much weaker second Josephson harmonics, bringing their contribution on par[25–31]. As a result, the Josephson potential energy, while remaining a $2\pi$-periodic function, acquires a more complex shape involving local minima, which can lead to much more anharmonic spectra. By contrast, higher harmonics are suppressed in conventional tunnel junctions, and hence flux bias merely rescales the regular Josephson potential, with weak effect on the anharmonicity[32]. So far, higher harmonics of the supercurrent has been mainly explored in the context of parity-protected qubits, which would emerge when the first harmonic is completely canceled. Achieving such cancellation, however, requires precise control of the junction parameters via electrical gating, which, in practice, has proven challenging[24,25,28]. In this paper, we describe a more immediate impact of the supercurrent interference in two transparent junctions by focusing on the half-flux quantum sweet spot, where the anharmonicity is already dramatically enhanced but the sensitivity to the 1/f flux noise is zero to first order. Such a device presents an opportunity to improve on the main drawback of transmons - the weak anharmonicity – using the unique property of mesoscopic superconducting-semiconducting junctions.

## Results

We investigate the anharmonic flux-tunable transmon qubits on an InAs quantum well (QW) proximitized by epitaxial Aluminum (epi-Al)[33,34] at the $\Phi = 0.5\Phi_0$ flux sweet spot. For simplicity we model the weak link JJs with their effective number of channel $N$ and a characteristic channel transparency $T$. The SQUID loop formed by two weak link JJs is shunted by a capacitance $C$ in between the superconducting island and the ground. The qubit potential is tuned by the applied magnetic flux $\Phi$ in the SQUID loop[35]. We denote the phase across the first JJ as $\varphi$ and the phase of the second JJ is $2\pi\Phi/\Phi_0 - \varphi$. As seen in Fig. 1a, the superconducting island is subject to voltage fluctuations modeled by a capacitively coupled electrode with an offset voltage $V_g$, which is equivalent to an offset charge of $n_g = C_g V_g/2e$[36]. The Hamiltonian of the qubit is given by the sum of the charging energy and the Josephson energy $U(\varphi)$:

$$H = 4E_C(n - n_g)^2 + U_J(\varphi),\qquad(2)$$

where $n$ and $\varphi$ are the number of the Cooper pairs and the phase operators, satisfying the commutator $[n, \varphi] = i$. The Josephson energy has a simple structure for the tunnel junctions, $U_J(\varphi) = E_J(1 - \cos\varphi)$. In a transmon with conventional AlOx JJs, the same scale determines the barrier height, the curvature and quartic terms at the potential minimum, resulting in competition between transmon sensitivity to charge noise and anharmonicity.

The current transport in a JJ with a semiconducting weak link is enabled by Andreev Bounds States (ABSs) formed by conduction channels in the weak link[37]. When the JJ has a superconducting phase bias $\varphi$, the energy of an ABS is given by

$$U(\varphi, T) = \Delta\sqrt{1 - T\sin^2\frac{\varphi}{2}},\qquad(3)$$

where $\Delta$ is the superconducting gap and $T$ is the conduction channel transparency. Owning to a finite lateral width in our JJs, multiple transverse channels contribute to the supercurrent across each JJ based on the lateral confinement. As such, the SQUID energy of a flux-tunable gatemon is determined by the sum over the contribution from all the ABSs in both the JJs:

$$U_{\text{SQUID}}(\varphi) = \sum_{k=1}^{M} U(\varphi, T_1^{(k)}) + \sum_{l=1}^{N} U(2\pi\Phi/\Phi_0 - \varphi, T_2^{(l)}),\qquad(4)$$

where $T_1^{(1)}, \ldots, T_1^{(M)}$ are the channel transparencies in the first JJ with $M$ total conduction channels and $T_2^{(1)}, \ldots, T_2^{(N)}$ are the channel transparencies in the second JJ with $N$ total conduction channels. This model

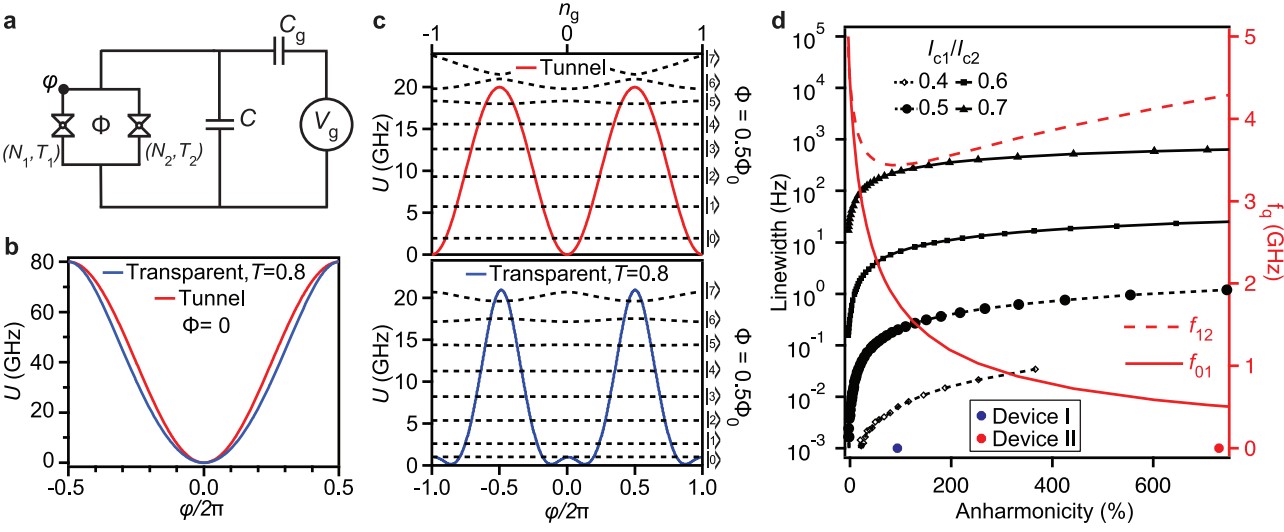

**Fig. 1 | Flux-tunable gatemon simulation. a** Schematics of the flux-tunable gatemon qubit comprising of two superconductor-semiconductor JJs in a SQUID loop, characterized by the number of channels $N$, channel transparency $T$, and shunted by a large capacitance $C$. **b-c** Josephson potential $U$ versus the JJ phase $\varphi$ at magnetic flux $\Phi = 0$ (**b**) and $\Phi = 0.5\,\Phi_0$ (**c**) for SQUID asymmetry $I_{c1}/I_{c2} = 0.6$. At $\Phi = 0.5\,\Phi_0$, the potential has a double-well shape due to the interference of Josephson energy harmonics in the SQUID. The black dashed lines represent the lowest-lying qubit energy states as a function of the offset charge $n_g$, modeled by a capacitance $C_g$ coupled to an offset voltage $V_g$ in (**a**). **d** Charge dispersion linewidth for different SQUID asymmetries $I_{c1}/I_{c2}$ (black curves) and the qubit transition frequencies for asymmetry $I_{c1}/I_{c2} = 0.7$ (red curves) as a function of the qubit anharmonicity at $\Phi = 0.5\,\Phi_0$. We experimentally achieve devices with anharmonicities of 96% and 730% with charge dispersion linewidths suppressed below 1 Hz, as marked by the blue and red solid circles respectively.

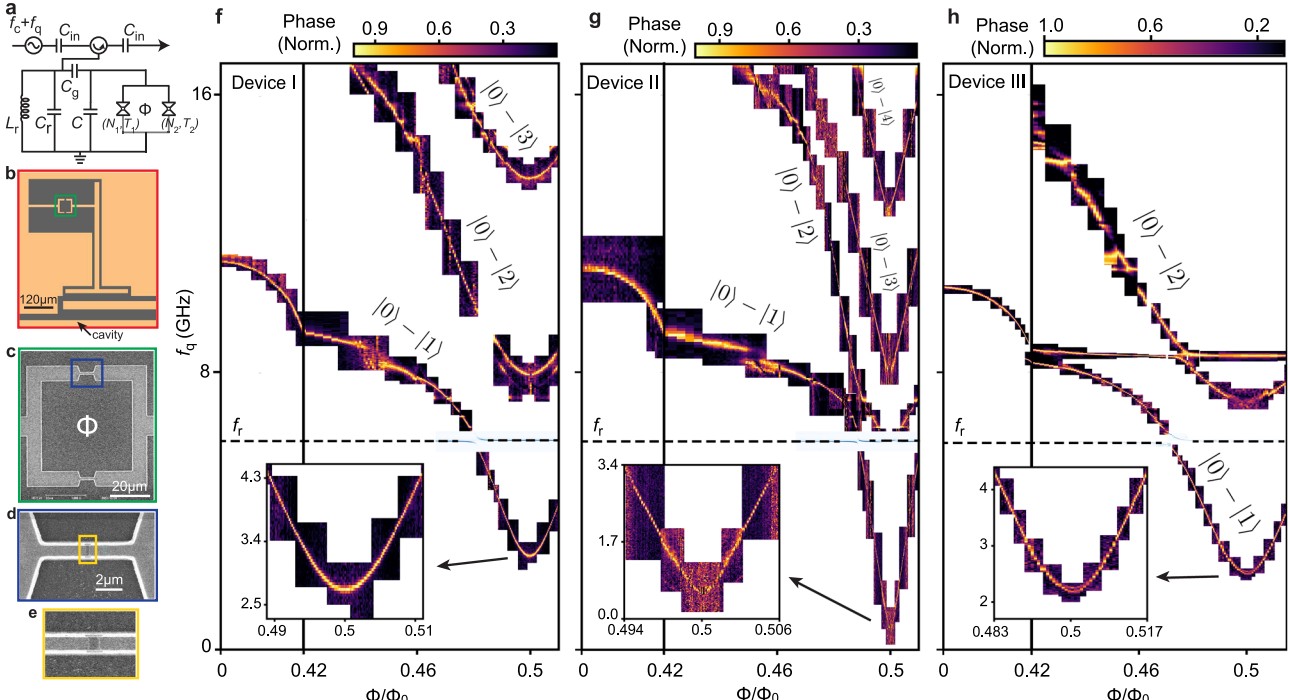

**Fig. 2 | Energy spectroscopy. a** Circuit diagram of the flux-tunable gatemon, capacitively coupled to a $\lambda/4$ resonator and embedded in a reflection measurement setup. **b–e** False-colored scanning electron microscopy images of the qubit, with zoomed-in images of the SQUID loop and the JJ. **f–h** Normalized phase response as a function of the qubit drive frequency $f_q$ and the applied external magnetic flux $\Phi/\Phi_0$ for device I (**f**), device II (**g**) and device III (**h**), respectively. The black dashed line denotes the resonator frequency $f_r$ as a function of $\Phi$. **Inset:** Zoomed-in plot of the $|0\rangle - |1\rangle$ qubit transition at $\Phi = 0.5\,\Phi_0$, highlighting the flux 'sweet spot'.

yields a good fit to the spectra of flux-tunable gatemon qubits with InAs nanowire JJs when the number of channels in the JJs is small[26,27,38]. When the number of conduction channel is large, Eq. (4) can be approximated by grouping the channel transparencies into one or a few characteristic transparencies or by determining the leading Fourier harmonics of the Josephson potential energies. In this study, we also compare the approximation methods mentioned above for modeling anharmonic flux-tunable transmon qubits with InAs-based 2DEG weak-link JJs by examining the quality of their fits to the measured qubit spectra.

The flux-tunable gatemon qubit biased at $\Phi = 0$ is equivalent to the typical gatemon qubit with similar qubit potential to a transmon qubit, as shown in Fig. 1b. When biased at $\Phi = 0.5\Phi_0$, the bottom of the potential changes to a flattened double-well shape near the minima of the well, drastically increasing the qubit anharmonicity. The double-well shape of the qubit potential arises from the destructive interference of the $\cos\varphi$ terms and constructive interference of the $\cos(2\varphi)$ terms in the two JJs' potentials. The qubit states see a high potential barrier that suppresses the tunneling probability to adjacent wells. Meanwhile, the flux-tunable gatemon potential has reduced curvature of the potential near its minima so the qubit has a large anharmonicity up to a few hundred percent. As seen in Fig. 1c, the simulated flux-tunable gatemon $|0\rangle$ and $|1\rangle$ states are insensitive to the offset charge, but the qubit is strongly anharmonic.

Although the trade-off between the charge dispersion and the anharmonicity still applies to the anharmonic tunable transmon qubit, the charge dispersion can be further suppressed by engineering the asymmetry of the SQUID. As shown in Fig. 1d, as the qubit anharmonicity increases, the charge dispersion linewidth increases by one to two orders of magnitude from a conventional transmon with <5% anharmonicity. The loss in the suppression of charge dispersion can be compensated with a more asymmetric design of the SQUID, which is equivalent to increasing the first Josephson harmonic in the SQUID current-phase relation (CPR). The charge dispersion sees four orders

of magnitude suppression going from $I_{c1}/I_{c2} = 0.7$ to $I_{c1}/I_{c2} = 0.4$, while the anharmonicity >100% remains achievable. The ability to simultaneously achieve suppressed charge dispersion and large anharmonicity for flux-tunable gatemon devices at the $\Phi = 0.5\Phi_0$ flux sweet spot promises better qubit performance than the conventional transmon counterpart.

An equivalent circuit diagram and false color scanning electron microscopy images of the investigated anharmonic flux-tunable transmon device are shown in Fig. 2a-e. We measure three nominally identical devices, all of which show similar spectra. Our typical device consists of a $\lambda/4$ resonator capacitively coupled to a common transmission line for microwave control and readout. A T-shaped Al island etched into the surrounding ground plane on an InP substrate provides the qubit shunt capacitance, which has an estimated charging energy $E_C \sim 180$ MHz from electrostatic simulations. The qubit, comprising of two weak link JJs in a SQUID loop, is connected to the T-shaped island and the ground plane via Al leads. Each individual JJ is formed by etching away a $L \sim 250$ nm long and $W \sim 800$ nm wide segment of the epi-Al. The qubit is read out using the $\lambda/4$ resonator with a resonance frequency of $f_r = 6.01$ GHz and Q-factor $\sim 400$, and measured in a dilution refrigerator at <50 mK using standard complex reflectance measurements, as shown in detail schematically in supplementary information 1.

We begin by measuring the normalized phase response of the readout resonator as a function of the magnetic flux $\Phi$, while sweeping the resonator drive frequency $f_r$. The resonator response for each device, indicated by the black dashed lines in Fig. 2f-h and shown in detail in supplementary information 2, exhibits a vacuum Rabi splitting with a cavity-qubit coupling strength of $g = 122$ MHz for device I, $g = 101$ MHz for device II and $g = 170$ MHz for device III, as shown in supplementary information 8. These values scale with the respective charge matrix elements and are indicative of qubit states hybridizing with the readout resonator mode. To further probe the qubit transitions directly, we perform two-tone spectroscopy, where the readout

**Table 1 | Fit parameters: channel transparency $T$, number of conduction channels $N$ and charging energy $E_C$, along with the corresponding 95% confidence intervals, determined from the single transparency model using Eq. (2) and (4)**

|  | $N_1\Delta$ (GHz) | $T_1$ | $N_2\Delta$ (GHz) | $T_2$ | $E_C$ (GHz) | $\eta$ |
|---|---|---|---|---|---|---|
| I | 221.02 ± 22.9 | 0.999 ± 0.099 | 98.90 ± 2.9 | 0.926 ± 0.006 | 0.197 ± 0.00 | 0.96 |
| II | 138.03 ± 3.0 | 0.928 ± 0.006 | 466.98 ± 88.5 | 0.542 ± 0.095 | 0.163 ± 0.002 | 7.3 |
| III | 52.03 | 0.968 | 806.25 | 0.147 | 0.350 | 1.25 |

The anharmonicity $\eta$ is extracted from measurements in Fig. 2f-h. The details of the error analysis are described in supplementary information 9.

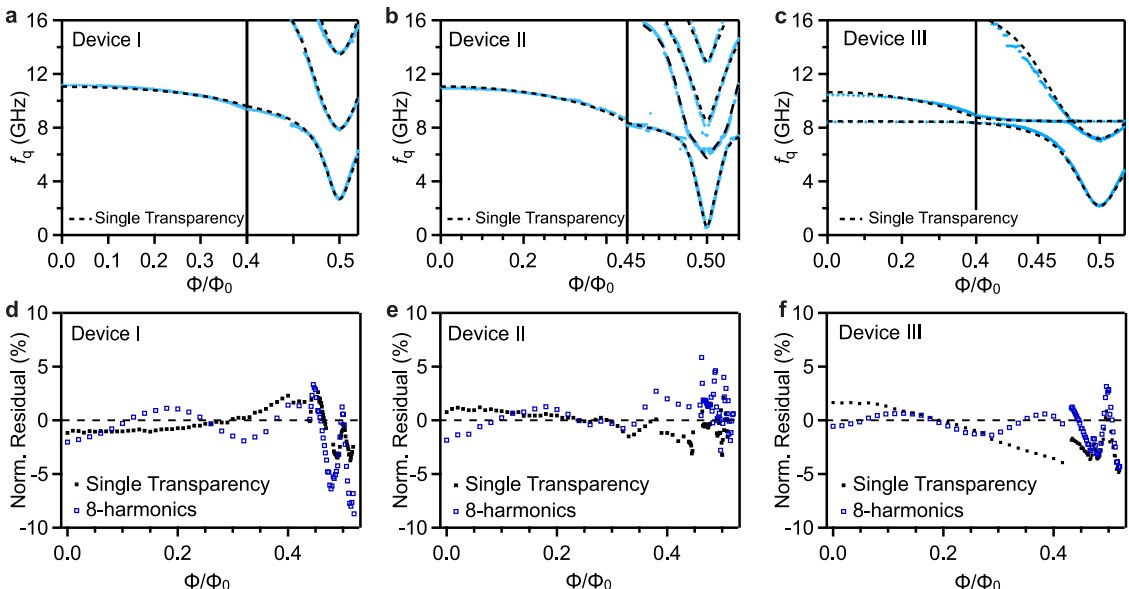

**Fig. 3 | Single transparency model fit and residual. a–c** Qubit transition spectrum as a function of the qubit frequency $f_q$ and the applied external magnetic flux $\Phi/\Phi_0$ for device I (**a**), device II (**b**) and device III (**c**) extracted from Fig. 2f, g and h, respectively. The dashed black lines show fits to the qubit transitions obtained from the Hamiltonian in Eq. (2) using the single transparency model. **d–f** Normalized residual $(f_{model} - f_{meas})/f_{meas}$ as a function of $\Phi/\Phi_0$ for device I (**d**), device II (**e**) and device III (**f**), respectively.

resonator drive tone is fixed at its resonant frequency for each flux, while a second drive tone $f_q$ is applied to excite the qubit transitions. When biased at $\Phi = 0$, each device in Fig. 2f-h shows a single qubit transition with transmon-like behavior, characterized by a decreasing $|0\rangle - |1\rangle$ transition frequency $f_{01}$ as $\Phi$ increases. In contrast, at $\Phi = 0.5\,\Phi_0$, we observe a rich spectrum featuring multiple flux-dependent transitions that strongly anticross symmetrically around $0.5\,\Phi_0$. The qubit anharmonicity shows a strong increase with increasing flux, changing from traditional gatemon and transmon-like values of $\eta \simeq 2\%$ at $\Phi = 0$ to $\eta \simeq 96\%$ for device I, $\eta \simeq 730\%$ for device II, and $\eta \simeq 125\%$ for device III at $\Phi = 0.5\,\Phi_0$ (see Table 1). In addition, the $|0\rangle - |1\rangle$ transition shows a first-order insensitivity to flux noise at $\Phi = 0.5\,\Phi_0$ (details in supplementary information 3), consistent with a flux 'sweet spot' at half-flux quanta.

In order to understand the observed spectrum, we simulate the measured qubit transitions by determining the eigenenergies of the qubit Hamiltonian in Eq. (2). The qubit transition frequencies extracted from Fig. 2f-h are shown in Fig. 3a-c with blue solid curves. To account for the large number of transparent conduction channels in superconducting-semiconducting JJs, we model the Josephson potential by a single characteristic channel transparency for each JJ. This approach is justified for atomically clean JJs, where channel-to-channel variations are expected to be minimal. However, in practice, only a fraction of the total channels ($\sim 10\%$ in our devices) is highly transmissive due to mesoscopic imperfections such as disorder and selective mode coupling[34,39–41]. Under this assumption, the Josephson energy $U_{SQUID}(\varphi)$ in Eq. (4) simplifies to $U_{SQUID}(\varphi) = N_1 U(\varphi, T_1) + N_2 U(2\pi\Phi/\Phi_0 - \varphi, T_2)$, where $T_1$ and $T_2$ are the channel transparencies of the first

and second JJ with $N_1$ and $N_2$ conduction channels, respectively. The black dashed curves in Fig. 3a-c show the calculated qubit transition frequencies with the fit parameters presented in Table 1, consistent with the short-junction limit and quasi-ballistic regime estimated in supplementary information 11 for our devices. In addition, our model indicates a strong suppression of the charge matrix element, as shown in supplementary information 4, for the $|0\rangle - |1\rangle$ qubit transition at $0.5\,\Phi_0$. This suppression diminishes the dipole coupling of the qubit to charge noise and dielectric loss channels, thereby enhancing the $T_1$ relaxation time at the half-flux sweet spot.

To quantitatively assess the accuracy of the single transparency model, we plot the normalized residual $(f_{model} - f_{meas})/f_{meas}$ vs $\Phi$ for the lowest-energy $|0\rangle - |1\rangle$ transition in Fig. 3d-f. As shown by the black filled squares, we observe a residual of $< 2\%$ at $\Phi = 0$ and no more than 4% at $\Phi = 0.5\,\Phi_0$ for device I and II, while device III remains below 5% for all $\Phi$. For comparison, we consider a higher-harmonic model in which the Josephson potential energy for each JJ is approximated by its leading Fourier harmonic contributions: $U_i = \sum_k E_{Ji}^k \cos(k\varphi)$, where $i \in [1, 2]$ refers to the first and second JJ. The normalized residual, determined using the four leading harmonic terms $k = 4$ (see supplementary information 5 for $k = 1$ and $k = 2$) for each JJ, exhibits an oscillating behavior with residual $\leq 10\%$ for device I and $\leq 5\%$ for devices II and III (blue hollow squares). These results indicate that the single transparency model provides the most accurate representation of the flux-tunable gatemon spectra despite using fewer fitting parameters. We observe that further increasing the number of characteristic transparencies does not significantly reduce the residual, as demonstrated in supplementary information 6.

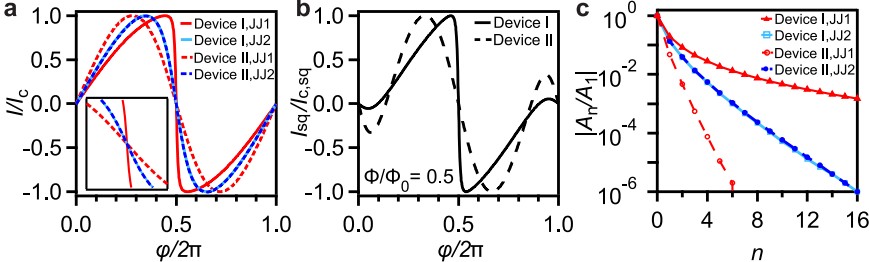

**Fig. 4 | Current phase relation (CPR). a** Current phase relation ($I/I_c$ vs $\varphi$) of each JJ in device I and II, determined using the fit parameters obtained from the single transparency model in Table 1. We denote the JJ with larger critical current as JJ1 in each device. Inset: Zoomed-in image of the CPRs at $\varphi/2\pi = 0.5$. **b** CPR of the SQUID loop ($I_{sq}/I_{c,sq}$ vs $\varphi$) at $\Phi = 0.5\,\Phi_0$ calculated using Eq. (5). **c** Normalized $n^{th}$-order harmonic contribution $|A_n/A_1|$ for the leading 16 terms, i.e. up to $n = 16$, with an amplitude baseline threshold set at $|A_n/A_1| = 10^{-6}$.

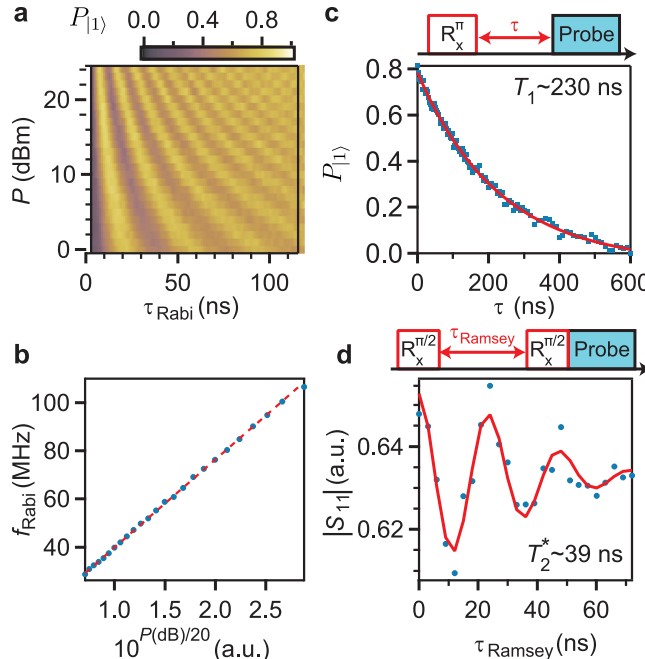

**Fig. 5 | Coherent qubit control and manipulation. a** Rabi oscillations as a function of the qubit drive power $P$ and the Rabi pulse duration $\tau_{Rabi}$ at $0.5\,\Phi_0$. **b** Rabi frequency $f_{Rabi}$ extracted from (**a**) as a function of $10^{P(dB)/20}$, which is proportional to the qubit drive amplitude $\sqrt{P}$. **c** Energy relaxation time $T_1$ measured at $\Phi = 0.5\,\Phi_0$. The solid red curve is an exponential fit with time constant $T_1 \sim 230$ ns. **d** Ramsey coherence time $T_2^* \sim 39$ ns measured at $\Phi = 0.5\,\Phi_0$.

The extracted fit parameters allows us to determine the CPR of the individual JJs and the SQUID loop. For each JJ, we use the Josephson potential: $I_i(\varphi_i) = \frac{2\pi}{\Phi_0}\frac{\partial U_i}{\partial \varphi_i}$ with $i \in [1, 2]$ for JJ1 and JJ2 respectively, to extract the normalized CPR $I/I_c$ for devices I and II in Fig. 4a. We observe highly skewed non-sinusoidal CPRs, reminiscent of higher-order harmonic contributions due to the interference of supercurrents carried by 2-electrons, 4-electrons and higher charges across the JJs. Consequently, for the SQUID loop CPR at $\Phi = 0.5\,\Phi_0$, we sum over the individual JJ contributions to obtain:

$$I_{sq}(\varphi) = I_1(\varphi) + I_2\left(2\pi\frac{\Phi}{\Phi_0} - \varphi\right) \qquad (5)$$

$I_{sq}/I_{c,sq}$ extracted for devices I and II is shown in Fig. 4b. We further extract the higher-order harmonic contributions for each JJ's CPR using: $I_i(\varphi_i) = \sum_n A_n \sin(n\varphi)$. Figure 4c shows the normalized $n^{th}$-order harmonic term $|A_n/A_1|$ for devices I and II, determined up to the leading sixteen terms ($n = 16$) with an amplitude baseline threshold

set at $|A_n/A_1| = 10^{-6}$. We observe a gradual fall-off in the harmonic amplitudes with increasing $n$. In addition, while some JJs exhibit similar CPRs, others show substantial deviations, resulting in distinct harmonic content. This diversity reflects the sensitivity of the CPR to microscopic transport parameters of the JJs. The harmonic analysis provides complementary information to standard transport measurements, allowing one to identify deviations from purely sinusoidal behavior and engineer the Josephson potential landscape including the realization of multi-well potentials, thereby tailoring qubit properties for novel circuit architectures and optimized performance.

We now demonstrate the basic operations of the flux-tunable gatemon qubit by performing Rabi oscillation measurements and Ramsey interferometry using time-domain manipulation and read-out. For each measurement, microwave pulses with frequency $f_q \approx f_{01}$ are applied via the cavity readout feedline, after an initialization time of $100\,\mu$s to allow the qubit to relax to the ground state $|0\rangle$. For Rabi measurements, we apply the drive pulse for a time duration $\tau_{Rabi}$, followed by reading out the qubit state via the cavity. The microwave pulse induces Rabi oscillations between the qubit states $|0\rangle$ and $|1\rangle$, as shown in Fig. 5a, where we plot the probability of qubit state $|1\rangle$, $P_{|1\rangle}$ as a function of $\tau_{Rabi}$ and the applied drive power $P$. We observe that the Rabi frequency increases for larger drive power $P$. To explicitly demonstrate this, we extract the Rabi frequency $f_{Rabi}$ for each power $P$ by fitting the Rabi oscillations in Fig. 5a with a sinusoidal function with exponentially decaying envelope: $A\sin(2\pi f_{Rabi}t)e^{-t/\tau_{Rabi}}$. The extracted $f_{Rabi}$ as a function of $10^{P(dBm)/20}$, which is proportional to the drive amplitude $\sqrt{P}$, shows a linear dependence in Fig. 5b, with Rabi frequencies as large as $>100$ MHz enabling coherent manipulation and gate operations of the flux-tunable gatemon qubit much faster than traditional transmons and gatemons.

We first measure the energy relaxation time $T_1$ to estimate the qubit coherence times. The Rabi oscillations data in Fig. 5a allow us to calibrate the pulse amplitudes and durations for the corresponding rotations around the $x$-axis on the Bloch sphere. To measure $T_1$, we first apply a $R_X^\pi$ pulse to excite the qubit to state $|1\rangle$, followed by a waiting time delay $\tau$ before readout. The qubit probability $P_{|1\rangle}$, plotted in Fig. 5c, as a function of the time delay $\tau$ shows an exponential decay due to the qubit relaxation, yielding $T_1 \sim 230$ ns for device I. Similarly, to measure Ramsey dephasing time $T_2^*$, two $R_X^{\pi/2}$ pulses slightly detuned from the qubit frequency, $\delta f \sim 40$ MHz and separated by a time delay $\tau_{Ramsey}$ are applied before readout. The resulting cavity readout response as a function of the time delay $\tau_{Ramsey}$, shown in Fig. 5d, demonstrates Ramsey fringes, consistent with the qubit state acquiring a phase $\varphi = 2\pi\delta f\tau_{Ramsey}$ while precessing around the $z$-axis of the Bloch sphere. We obtain $T_2^* \sim 39$ ns by fitting to a sinusoidal function with an exponential decay envelope.

## Discussion

As a last step, we critically evaluate the dominant loss mechanisms affecting the qubit's relaxation and decoherence times. We observe that $T_1$ is independent of the qubit frequency $f_{01}$ in the vicinity of $\Phi = 0.5\,\Phi$ (shown in Supplementary Fig. 7), which we then use to determine the effective dielectric loss tangent $\tan\delta_c$ using the charge matrix elements from supplementary information 4. We obtain $\tan\delta_c$ to be on the order of $\sim 10^{-4}$, consistent with typical values for dielectrics such as InP and AlOx, but significantly higher than the typical values of $\sim 10^{-6}$ observed for traditional transmon devices on silicon and sapphire substrates. This suggests that dielectric losses, likely originating from the InP host substrate, limit $T_1$. Additionally, we note that the dephasing time $T_2 \ll 2T_1$, indicating that qubit coherence is not primarily limited by energy relaxation, but rather by other on-chip dephasing mechanisms, for example, by $1/f$ noise from critical current fluctuations[16] or the decoherence of the Andreev Bound States close to the $\varphi = \pi$ phase bias[42].

In conclusion, we demonstrate a hybrid InAs-Al flux-tunable transmon qubit biased at the half-integer flux sweet spot, which exhibits a significant enhancement in anharmonicity - over an order of magnitude increase - without any electrical gating, along with a minimal charge dispersion, thereby reducing its sensitivity to offset charge noise. Additionally in our model, we observe a suppressed charge matrix element by a factor of 2, enabling partial parity protection of the qubit without the need for strict symmetry between the two JJs'. This eliminates the need for precise fabrication with accurate electrical gating or additional circuit elements[28,30] necessary to maintain the symmetry. Such large intrinsic qubit anharmonicities enable fast single-gate times of <10 ns, surpassing the Rabi frequencies typically achieved in traditional gatemons and transmons without any pulse shaping. Although dielectric losses currently limit the qubit lifetime, the use of less lossy substrate materials, for example, Si or sapphire, should make it possible to achieve much longer coherence and energy relaxation times. Our work is versatile enough to be implemented in other material systems such as Ge or SiGe, and provides a novel approach to characterize various superconducting-semiconducting material platforms and implement strongly anharmonic, partially protected qubits for quantum information processing without the need for complex design and fabrication.

## Methods

The anharmonic tunable transmon qubit device is fabricated on a stacked heterostructure grown on $500\,\mu m$ thick Fe-doped InP substrate. An $In_{0.53}Al_{0.47}/In_{0.52}Al_{0.48}$ superlattice and a graded $In_xAl_{1-x}As$ layer are further grown on the substrate to act as a buffer layer, followed by 4nm of InGaAs, 4 nm of InAs and 10 nm of InGaAs, which defines the quantum well structure with InAs as the 2DEG layer. The wafer is then capped with a $25 - 30$ nm thick layer of Al deposited $in - situ$ with molecular beam epitaxy.

The coplanar waveguide (CPW) resonators are fabricated on a 6 mm by 5 mm rectangular wafer with a center conductor width of $35\,\mu m$ and a trench gap width of $20\,\mu m$ using e-beam lithography. The capping Al layer is then removed with 50℃ Transene Aluminum Etchant Type D, followed by a wet-etch of the III-V layers at room-temperature for 3-4 min, resulting in a trench depth of $350 - 450$ nm. The III-V layer etchant is a mixture of phosphoric acid (85%), hydrogen peroxide (30%) and DI water with a volumetric ratio of 1:2:60 respectively. The qubit SQUID, fabricated in the same step as the CPW resonators, has a dimension of $50\,\mu m$ by $50\,\mu m$ for devices 1 and 2, and $22\,\mu m$ by $22\,\mu m$ for device 3. We then etch the Al for 6 seconds in a separate e-beam lithography step to form the Josephson junctions. The device is finally packaged in a copper cavity via Al bond wires and measured in a dilution refrigeration at $< 50$ mK using standard reflectance measurements.

## Data availability

All data in this publication are available in numerical form at https://doi.org/10.5281/zenodo.17654710.

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

## Acknowledgements

V.E.M. has received funding for this work from ARO NextNEQST (contract no. W911-NF22-10048) program. J.S. acknowledges support from ONR MURI (award no. N00014-22-1-2764) and ARO NextNEQST grant mentioned above.

## Author contributions

S.L. and A.B. fabricated the devices, performed the measurements, analyzed and interpreted the data and simulated the theoretical model. M.G.V. provided theoretical support in simulating the model. J.I., I.L. and J.S. have grown the InAs/Al 2DEG material. V.E.M. helped in interpreting the data. A.B. and S.L. wrote the paper with inputs from all the authors. V.E.M. initiated and supervised the project. All authors discussed the results and contributed to the manuscript.

## Competing interests

The authors declare no competing interests.
