## [Transparent Peer Review file · Nature Communications]

Strongly anharmonic flux-tunable transmon based on InAs-Al 2D heterostructure

Corresponding Author: Dr Arunav Bordoloi

Version 0:

Reviewer comments:

Reviewer #1

(Remarks to the Author)

This paper presents a detailed study of flux-tunable hybrid Josephson junction qubits, demonstrating strong anharmonicity without the need for electrostatic gating. Using spectroscopic techniques, the authors extract the current-phase relation (CPR) of the hybrid junctions and showcase coherent qubit control at the half-flux sweet spot, including the observation of fast Rabi oscillations. The findings on CPR are particularly relevant to the superconducting-semiconducting hybrid community. While the pronounced anharmonicity is a key advantage for qubit control and gate fidelity, its potential impact is tempered by relatively limited coherence times, which may constrain the practical utility of these qubits.

I find that there is a significant issue with a main claim of the paper. The authors state that the gate time they have achieved using their qubit (≤ 10 ns) is significantly faster than existing transmons, which is not true. The typical anharmonicity of a transmon is ~ 250 MHz, which sets the speed limit for when leakage to non-computational states will occur. While it is true that Rabi rates above 100 MHz in AlOx-based transmons will exhibit leakage, multiple groups have used the DRAG protocol to make sub-10 ns gates in standard AlOx-based transmons. I cannot support publication of this paper until the authors clarify their claim regarding their gate time relative to existing transmons.

The following comments should also be addressed in their manuscript:

1. Since the authors do not use any electrostatic gating in their hybrid superconductor-semiconductor Josephson junctions, the term “gateless gatemon” appears somewhat redundant. Referring to the device simply as a flux-tunable transmon with hybrid junctions more accurately and clearly reflects the nature of the work.
2. In Figure 1(d), the measured anharmonicity for devices 1 and 2 is marked with dots. However, given the already dense presentation of the graph, these indicators are difficult to understand as data and may be misinterpreted as markers.
3. In Figure 2(h), a transition is visible above the 0–1 transition and below the 0–2 transition, but the authors do not provide any explanation for this feature. Clarifying its origin would be helpful.
4. The authors model the Andreev bound state (ABS) energy using the short junction approximation. Given the junction length of 250 nm, which is relatively long, have the authors considered alternative models beyond this approximation? As highlighted in Fatemi et al. (SciPost Phys. 18, 091), such junctions may deviate significantly from the short junction regime.
5. The authors should also provide additional motivation for why they expect all of the channels in their junction to have the same transparency.
6. The authors could provide a discussion on the importance of the higher-order harmonic contributions to the current. I assume a numerical derivative is performed on the potentials that they found to get the current-phase relation. The extracted current-phase relation is skewed so the authors then expand it in a sine series and find the higher-order harmonic contributions. It would be useful to understand how these higher-order contributions could provide information about the junction or guidance for future designs that was not previously available.
7. Have the authors performed T2E (echo coherence time) versus flux measurements? Such data would provide further insight into dephasing mechanisms and flux noise sensitivity in these devices.
8. The conclusion mentions suppressed charge matrix elements as a key result, but this point is only addressed in the supplementary material. Given its importance, incorporating a brief discussion in the main text would strengthen the narrative and highlight its relevance to qubit coherence. It would also be useful to plot the expected T1 times vs flux for the quality factor quoted in the text in the same plot as their measured T1 times.

Reviewer #2

(Remarks to the Author)

The manuscript reports on a implementation of gatemon qubits using split-junction devices based on InAs/Al 2D heterostructures. The key advance is the demonstration of significantly enhanced anharmonicity (>100%) at the half-integer flux sweet spot, enabling fast and potentially more robust gate operations without requiring gate voltages. The authors also use high-resolution spectroscopy to extract detailed features of the current-phase relation, offering insight beyond standard transport techniques.

The results are timely and relevant to the community working on hybrid superconducting-semiconducting qubits and may present a significant step toward leveraging transparent junctions in scalable quantum hardware. However, there are several points that should be addressed before the manuscript can be considered for publication.

Comments:

- The phrase "anharmonicity in excess of 100%" in the abstract is confusing. Anharmonicity is defined formally in Eq. (4), but the term is used repeatedly before that point in the manuscript. The authors should clarify this early in the text, preferably when the concept is first introduced. For instance, is 100% referring to the qubit frequency itself, or relative to the fundamental harmonic level spacing?
- In Fig. 1(d), the mapping between plotted lines and the left/right y-axes is unclear. The caption and figure labeling should be revised to indicate clearly which data correspond to which axis. As it stands, this figure is difficult to interpret.
- The transition from Eq. (2) to Eq. (3) introduces the concept of multiple channels in the Josephson junction, but the text does not explain this development adequately. Please expand this section to clarify the physical picture, assumptions, and implications of including multiple conduction channels.
- The coupling strength g is introduced in connection with the spectroscopy data, but its precise definition is missing. This is crucial, as multiple conventions exist in the literature. For example, whether the Hamiltonian term is written as $\hbar g(\sigma_- + \sigma_+)$ or with $\hbar g/2$, or hg etc which all leads to numerical differences. Please state the exact convention used.
- Furthermore, how is g extracted? Is it taken directly from the vacuum Rabi splitting at resonance, or obtained via a full fit of the qubit-resonator avoided crossing as a function of flux? For devices with broad linewidths, a full fit generally provides a more reliable estimate, and this should be clarified.
- The extracted values of g differ significantly between the three devices, despite nominally identical qubit designs and similar resonator frequencies. The authors should comment on this variation. Is it due to fabrication variation, junction asymmetry, or some other source? If it is due to the different matrix elements from the various junction transparencies, then it should be mentioned clearly.
- There appears to be a missing T_2 in the inline equation before Table 1, which makes the expression incomplete. Please correct this.
- The manuscript claims suppression of the charge matrix element near the sweet spot. However, this does not seem to be supported by experimental evidence in the current work. In the supplemental material, a theoretical calculation is shown, but the flatness of T_1 -times across the sweet spot implies no observable suppression. Do the authors see any change in Rabi frequency that might indicate such suppression? If so, they should quantify this, but note that such a measurement would require a careful calibration of the entire drive line including PCB and device response. Without this, the claim should be moderated.

The work presents interesting and potentially impactful experimental results that merit publication in Nature Communications after the above issues have been addressed. I recommend revision to clarify the points above.

Reviewer #3

(Remarks to the Author)

Reviewer #4

(Remarks to the Author)

The authors of the manuscript "Liu et al., Strongly-anharmonic gateless gatemon qubits based on InAs/Al 2D heterostructure," study the properties of transmon devices in which the qubit Josephson junction has been replaced by an asymmetric DC SQUID with semiconductor Josephson junctions (Sm-JJs). The Sm-JJs are characterized by a current-phase relation (CPR) that contains higher harmonics in phase, ϕ . When the SQUID flux is fixed at half of a flux quantum

($\Phi_0/2$), the supercurrent interference partially cancels the $\cos \phi$ -component of the SQUID energy-phase relation. This results in a modified transmon spectrum with increased anharmonicity while, at the same time, retaining an acceptable charge dispersion linewidth. The authors report measurements of the qubit dispersion as a function of SQUID flux on three different devices. Next, they fit the transmon spectra, replacing the Josephson energy term by an ansatz for the flux/phase-dependent energy of a SQUID with two junctions. They compare two different approaches. The first one uses a parametrization of the junction energy-phase relation that assumes N_i identical channels of transparency T_i and gap Δ in the short channel limit (following Beenakker, Ref 37). The other model constitutes of a series of $\cos \phi$ -harmonics up to eighth order. Both models yield fits of roughly similar accuracy. The authors proceed to plot the SQUID CPR and the CPRs of the individual junctions that correspond to the extracted model parameter. Finally, they report the results of the qubit characterization for SQUID flux $\Phi_0/2$, extracting the characteristic decoherence times T_1 and T_2^* .

The work addresses one of the current big topics in the field of solid-state quantum information processing: the hunt for qubit systems with long coherence times and short gate operation times. The superconducting transmon qubit has the advantage of reduced charge sensitivity, but its small spectral anharmonicity increases the time-scale of gate manipulations. In this work, the authors make use of higher harmonics in the potential of a SQUID with Sm-JJs to engineer a qubit spectrum with larger anharmonicity. Whereas the idea of using higher harmonics is not entirely new, the authors present a neat demonstration of the concept that will be of interest to the wider community.

I read the manuscript with great interest but think that the presentation and analysis lack depth. Additionally, important information is missing, which precludes the reader from reaching their own conclusions. In its current state, I cannot recommend the paper for publication in Nature Communications.

Let me elaborate:

The data in Tab. 1 are not discussed.

The choice of parametrization of the junction potential $U(\phi, T)$, implies that the supercurrent transport can be modeled as identical channels in the short junction limit. Taking a look at the results of the fit (Tab. 1), I am wondering if this makes sense. I am particularly intrigued by the transparency values, $T \sim 0.928-0.999$. These are incredibly large values for semiconductor Josephson junctions, for which one expects a sizable mismatch between the Fermi surfaces of semiconductor and metal, consequently interface scattering and lower transparencies.

- Have the authors critically accessed the interpretation of highly transmitting channels?
- Has the interface quality been studied in similar JJs? (References?)
- Are the Sm-JJs indeed in the short junction limit? I.e., $L \ll \xi \approx \hbar v_F / 2\Delta$? ($L \equiv$ electrode separation; $v_F \equiv$ Fermi velocity in the semiconductor)

I didn't find the value of Δ in the text. Has it been measured or inferred from the critical temperature of the Al layer?

- Do the values N_i match expectations based on lateral size quantization?
- It is mentioned that all Sm-JJs are nominally identical. Why is there a large spread in the parameter values of Tab. 1? What are the values of the critical current for each junction?
- What is the mean free path in the semiconductor material? Can the authors rule out the diffusive transport limit?

I recommend the authors to provide this information in the revised manuscript and include references to available literature.

There are several aspects of the CPR discussion (page 12f and Fig.4) that I find strange:

In Fig. 4, the normalized CPRs of JJ2 of Device I and II match. They are calculated based on the fit parameters in Table 1. However, judging by the transparency values given there, I would expect JJ1 of Device II to match with JJ2 of Device I. Are the labels in Fig. 4 wrong, or am I missing something in the analysis? Is there a way to know which one of the two physical junctions has the parameters of JJ1 and JJ2 in this experiment? Or are the labels interchangeable?

What do we learn from Fig. 4c? The authors provide a plot of the first 16 amplitudes of the Fourier series expansion of eq. 2 for the parameters T_i in Tab. 1. In other words, an (effectively) one-parameter expression (as N_i and Δ drop out by normalization) is recast into another expression with 16 parameters. It is stated that the magnitudes of the amplitudes strongly decrease with increasing index n . This is true but easily understood when we look at the Fourier series of a sawtooth waveform, $|A_n| \propto 1/n$, which represents the extreme limit of a skewed CPR. I don't see the value of this analysis. On the other hand, in Fig. 3, the authors compare the single-transparency fit with a model for U_{SQUID} composed of eight $\cos \phi$ -harmonics. How do the coefficients of the Fourier series expansion (A_n) in Fig. 4c compare to the eight fitting parameters of the second model? Can we learn something about the junction physics or the fitting procedure by comparing the numbers?

I disagree with the statement "More notably, the differences in the CPR and harmonic contributions between individual JJs are minimal, enabling us to extract fine details in the current-phase relations, far surpassing the capabilities of any standard transport measurement."

The first part is wrong. It is observed that the CPRs of two junctions are identical. The other two are substantially different. Therefore their harmonic content is different. The second part is not discussed in the manuscript, and I highly doubt it is true. Excluding other systematic errors for now, the accuracy of the CPR extraction depends on the choice of a suitable parametrization and the quality of the fit to the experimental data. The authors have omitted a good deal of error analysis here. What are the confidence intervals for the parameters in Tab. 1? How large are the error bars in Fig. 4a,b? In Fig. 4c? Can the authors provide examples by detailed comparison with transport experiments in the literature to support their statement?

Finally, the authors should provide an analysis of systematic errors in their experiment. E.g., generally, the phase drop across the SQUID loop inductance must be taken into account when determining the CPR. If the inductance is large, the shape of the CPR has to be determined self-consistently. Have the authors calculated/estimated the loop inductance (geometric and kinetic components) and ascertained that it is safe to omit it from their analysis? How would it affect the precision of their CPR extraction and the fit values for transmission in Tab. 1?

Additional comments:

There is a reference missing on page 16, bottom line.

I, personally, find expressions like "gateless" gatemon or "super-semi" Josephson junctions confusing and distracting. The oxymoron in the title made me skip the paper when it first appeared on the arXiv. I presume other researchers feel similarly about colloquialisms.

The authors state that their transmon qubit allows for faster gate operations than traditional transmon and gatemons. As someone who works in the

periphery of the quantum information field, I would greatly appreciate if the authors gave more context and examples. It would benefit the general readership.

Version 1:

Reviewer comments:

Reviewer #1

(Remarks to the Author)

The authors have adequately addressed the concerns and comments from the previous round of review.

I do have one significant issue with their new revision regarding the suppression of the charge matrix element from their dielectric loss model. Their procedure seems to be circular - they extract the dependence of the loss tangent at each frequency using their measured T1 and then claim good agreement between their model, which uses the extracted loss tangent, and their experimental T1. The typical method for comparing with a loss model is to consider the expected functional form of T1 for some loss model, which includes the matrix element and a well-motivated frequency-dependent form for the loss tangent, and check whether the data can be fit using that model. The authors need to either remove the claim that their model fits their data well or provide reasoning for why their procedure is physically sound.

Reviewer #2

(Remarks to the Author)

The authors have addressed the previously raised points from myself and the other referees and I recommend publication

Reviewer #3

(Remarks to the Author)

Reviewer #4

(Remarks to the Author)

The authors have addressed all points that needed clarification in their rebuttal letter. I noticed, however, the sentence "This approach is justified for atomically clean JJs, where channel-to-channel variations are minimal." has been added on page 11 of the revised manuscript. I think this statement is misleading. Firstly, in their rebuttal letter, the authors point out that the variations in the critical current of the individual junctions likely originate from disorder or some other mesoscopic imperfections. Secondly, only about 1/10 of ~110 modes are highly transmissive due to either selective coupling or some sort of disorder/mesoscopic imperfection. I recommend this information be stated clearly (including the references on selective coupling and mode transmissions provided in the rebuttal letter) for the benefit of the reader.

I support the publication of the paper in Nature Communications.

Version 2:

Reviewer comments:

Reviewer #1

(Remarks to the Author)

The authors have addressed my comments in the revised version of their manuscript.

Reviewer #3

(Remarks to the Author)

Referee's comments:

Referee #1 (Remarks to the Author):

This paper presents a detailed study of flux-tunable hybrid Josephson junction qubits, demonstrating strong anharmonicity without the need for electrostatic gating. Using spectroscopic techniques, the authors extract the current-phase relation (CPR) of the hybrid junctions and showcase coherent qubit control at the half-flux sweet spot, including the observation of fast Rabi oscillations. The findings on CPR are particularly relevant to the superconducting-semiconducting hybrid community. While the pronounced anharmonicity is a key advantage for qubit control and gate fidelity, its potential impact is tempered by relatively limited coherence times, which may constrain the practical utility of these qubits.

Our response:

We thank the referee for accurately summarizing our work.

Referee #1

I find that there is a significant issue with a main claim of the paper. The authors state that the gate time they have achieved using their qubit (≤ 10 ns) is significantly faster than existing transmons, which is not true. The typical anharmonicity of a transmon is ~ 250 MHz, which sets the speed limit for when leakage to non-computational states will occur. While it is true that Rabi rates above 100 MHz in AlOx-based transmons will exhibit leakage, multiple groups have used the DRAG protocol to make sub-10 ns gates in standard AlOx-based transmons. I cannot support publication of this paper until the authors clarify their claim regarding their gate time relative to existing transmons.

Our response:

We thank the referee for this helpful comment. We agree that the conventional transmons can achieve sub-10 ns single-qubit gates through precise calibration with the DRAG protocol [Motzoi *et al.*, PRL 103, 110501 (2009); Lucero *et al.*, Phys. Rev. A 82, 042339 (2010)]. We have revised our statement to avoid claiming a unique speed advantage, but rather emphasize that our devices exhibit large intrinsic anharmonicity which enables >100 MHz raw Rabi frequency without the need of any specialized DRAG-type pulse shaping and sequences. Such high Rabi frequency facilitates strong drives for fast gate operations, thereby simplifying control and calibration with fewer resources, while intrinsically reducing leakage.

To address the referee's concern, we include the following changes (marked in red, in the revised manuscript):

1. We now state that our devices enable *“fast gates with raw Rabi frequencies exceeding 100 MHz, without any pulse shaping, enabled by the large intrinsic anharmonicity of our devices.”*
2. We explicitly note that fast gates are achievable in both our devices and standard AlOx-based transmons, our architecture naturally reduces leakage through spectrum engineering, thereby simplifying experimental implementation and control.

Referee #1

The following comments should also be addressed in their manuscript:

1. Since the authors do not use any electrostatic gating in their hybrid superconductor-semiconductor Josephson junctions, the term “gateless gatemon” appears somewhat redundant. Referring to the device simply as a flux-tunable transmon with hybrid junctions more accurately and clearly reflects the nature of the work.

Our response:

We thank the referee for this comment. We have changed the term “gateless gatemon” to “flux-tunable transmon” in the revised manuscript.

Referee #1

2. In Figure 1(d), the measured anharmonicity for devices 1 and 2 is marked with dots. However, given the already dense presentation of the graph, these indicators are difficult to understand as data and may be misinterpreted as markers.

Our response:

We thank the referee for the suggestion to improve figure 1d. In the revised figure, we have made the correspondence between the plotted data and the left/right y-axis clearer. In particular, we have added a new legend and improved axis color-coding to unambiguously indicate which data belongs to a particular axis. The key device parameters, i.e. anharmonicity and charge dispersion linewidths which are suppressed below 1 Hz, are now directly reported in the figure caption for better comprehensibility.

Referee #1

3. In Figure 2(h), a transition is visible above the 0–1 transition and below the 0–2 transition, but the authors do not provide any explanation for this feature. Clarifying its origin would be helpful.

Our response:

We thank the referee for pointing this out. The observed transition corresponds to a cavity mode, which arises from the resonant frequency of the sample holder cavity. This cavity mode strongly couples to all modes in our sample, inducing a measurable response in the qubit’s readout through dispersive coupling. We note here that this mode is specific to the cavity used for device 3, whereas devices 1 and 2 were measured in a different, although similarly designed, cavity.

Referee #1

4. The authors model the Andreev bound state (ABS) energy using the short junction approximation. Given the junction length of 250 nm, which is relatively long, have the authors considered alternative models beyond this approximation? As highlighted in Fatemi et al. (SciPost Phys. 18, 091), such junctions may deviate significantly from the short junction regime.

Our response:

We thank the referee for this important question, as the validity of the short-junction approximation is indeed central to our modeling of the Andreev bound states (ABS) and the current-phase relation (CPR). In a superconductor-semiconductor-superconductor (S-Sm-S) junction, the short-junction limit is defined by:

$$L \ll \xi = \frac{\hbar v_F}{\pi \Delta}$$

where L is the JJ length, ξ is the superconducting coherence length, v_F is the Fermi velocity in the semiconductor and Δ is the induced superconducting gap. Using $\Delta \approx 230 \mu\text{eV}$ determined from the Al critical temperature of $T_c \sim 1.5\text{K}$ [Yuan *et al.*, J. Vac. Sci. Technol. A 39, 033407 (2021)] and $v_F \sim 10^6\text{m/s}$ corresponding to an electron density of $n_s \approx 7.68 \times 10^{11}\text{cm}^{-2}$, we estimate $\xi \sim 1 \mu\text{m}$. For our junction length $L = 250 \text{ nm}$, this gives $L/\xi \sim 0.25$, placing our devices in the crossover of the short and intermediate regime.

To explicitly determine the validity of our approximation, we compare the extracted CPRs to expectations from a finite-length scattering formalism as discussed in Fatemi *et al.* (SciPost Phys. 18, 091). In the long-junction limit, the CPR evolves toward a more linear, sawtooth-like form near $\Phi = \pi$ with suppressed higher harmonics. In contrast, our CPR consistently displays strong forward skewness, indicative of high transparency channels due to substantial higher-harmonic content – clear hallmarks of the short-junction regime. To further investigate this, we also employ an empirical harmonic expansion (up to 4 harmonics per JJ) to extract a model-independent CPR alongside the short-junction ansatz, where we find that both approaches show excellent agreement to experimental data, confirming that the essential skewness and anharmonicity are robust. This behavior is consistent with prior works on accurately capturing CPR features in highly transparent, ballistic InAs-Al JJs [e.g., Mayer *et al.*, Nat. Comm. 11, 212 (2020); Goffman *et al.*, New J. Phys. 19, 092002 (2017)], and our simplified single-transparency model yields excellent fits across a wide range of flux and frequency.

While more elaborate finite-length models could, in principle, be used, such models require detailed knowledge of the JJ potential and disorder profile, information inaccessible from our two-tone qubit spectroscopy alone and would introduce many unconstrained degrees of freedom with minimal additional constraint from our data. Therefore, the lack of any experimental signatures of long-junction behavior and the excellent agreement of our fits, we conclude that the short-junction approximation is justified and sufficient to describe the dominant physics of our devices.

Referee #1

5. The authors should also provide additional motivation for why they expect all of the channels in their junction to have the same transparency.

Our response:

We thank the referee for this insightful question. In our short-junction single transparency model, we initially considered a distribution of channel transparencies by dividing them into

bins of width 0.2 (e.g. 0-0.2, 0.2-0.4 etc.). For each bin, we then assigned a characteristic channel transparency, which is treated as a fit parameter. In practice, the fits consistently converge the population into a single high-transparency bin for each JJ, e.g. around 0.99 for the 0.8-1 bin. This motivated us to adopt a simplified single-characteristic transparency for each JJ. Such behavior is reasonable for our InAs-Al 2DEG junctions, where molecular beam epitaxy (MBE) growth yields highly uniform, atomically clean interfaces. The resultant suppression in the interface roughness and inhomogeneous disorder reduces channel-to-channel variations in the transmission probabilities, consistent with prior studies in similar systems [e.g., Mayer *et al.*, Nat. Comm. 11, 212 (2020); Goffman *et al.*, New J. Phys. 19, 092002 (2017)], which have reported junctions with a few dominant highly transparent channels.

To further critically assess our single-transparency assumption, we implemented a two-transparency model by introducing an additional characteristic transparency and its corresponding channel population as additional fit parameters for each JJ. The resulting fits show two closely spaced transparencies and only a marginal improvement compared to the single-characteristic transparency case, as shown by the normalized residual plots in supplementary figure 6. This suggests that an average single-transparency approximation already captures the dominant features of the qubit dispersion and extracted CPR. While it is likely that small variations may exist among individual channels, they are not resolvable within the sensitivity of our current spectroscopy and do not meaningfully improve the fits.

Referee #1

6. The authors could provide a discussion on the importance of the higher-order harmonic contributions to the current. I assume a numerical derivative is performed on the potentials that they found to get the current-phase relation. The extracted current-phase relation is skewed so the authors then expand it in a sine series and find the higher-order harmonic contributions. It would be useful to understand how these higher-order contributions could provide information about the junction or guidance for future designs that was not previously available.

Our response:

We thank the referee for this valuable suggestion. Indeed, the higher-order harmonic content of the current-phase relation (CPR) provides rich information about the microscopic transport properties of the JJ, the shape of the Josephson potential and ultimately the qubit performance. We have included a discussion on these aspects in the revised manuscript.

More specifically, **in transport measurements**, the relative amplitude of the harmonic terms A_n serve as direct evidence of the junction's microscopic transport regime. A large A_1 with negligible $A_{n>1}$ corresponds to a purely sinusoidal CPR, characteristic of tunnel junctions with low transparency and diffusive transport. In contrast, significant second and third harmonic terms reflect highly transparent, ballistic or quasi-ballistic modes, which skew the CPR forward. A slow decay of A_n with increasing n indicates short-junction behavior with near-unity transmission channels. In our device, we find $A_2/A_1 \approx 1/2$ for the

SQUID loop and $A_2/A_1 \approx 1/6$ for individual JJs, along with non-negligible higher harmonics ($n \geq 3$), confirming the presence of a small number of highly transparent channels.

In addition, **the Josephson potential landscape** is determined by the relation:

$$U(\phi) \propto - \sum_n \frac{A_n}{n} \cos(n\phi)$$

Here, higher harmonics reshape the potential beyond the simple cosine form – dominant $A_{n>1}$ terms create skewed or multi-well potential landscape within a single 2π phase period, producing multiple local energy minima. This directly affects the curvature of the potential and thus the qubit’s anharmonicity, charge dispersion and sensitivity to flux noise, as demonstrated in our work.

Finally, **for qubit design**, controlling the harmonic content provides a versatile route to engineer novel qubit modalities. For example, $\cos(2\phi)$ qubits can be realized when the first harmonic is suppressed ($A_1 = 0$), achieved in symmetric SQUIDs [Larsen et al., PRL 125, 056801 (2020)] where the Josephson potential becomes π -periodic, i.e. a double well potential, with degenerate ground states, leading to an increase in T_1 at the half-flux sweet spot. Similarly, “harmonium” qubits [Hays et al., arxiv:2502.15459 (2025)] exploit the regime $A_1 \ll A_2$ with $A_2 < 0$, leading to non-degenerate but noise resilient ground states.

In summary, the higher-order harmonics are not just mathematical artifacts but a powerful diagnostic and design tool that encode real information - they provide a direct probe of the microscopic transport regime, determine the Josephson potential landscape, and enable harmonic engineering for optimized qubit performance and novel qubit architectures.

Referee #1

7. Have the authors performed T2E (echo coherence time) versus flux measurements? Such data would provide further insight into dephasing mechanisms and flux noise sensitivity in these devices.

Our response:

We thank the referee for pointing this out. Yes, we have performed T2E (Hahn echo coherence time) as a function of flux and qubit frequency around $0.5\Phi_0$, which are shown below:

We observe that T2E is independent of the flux around $0.5\Phi_0$, even though $\partial f_{qubit}/\partial\Phi$ increases away from the half-flux sweet spot, as shown in supplementary figure 3. This indicates that $1/f$ type flux noise is not the dominant dephasing mechanism in our qubit. Since $T_{2E} < 2T_1$, we note that other dephasing mechanisms such as critical current fluctuations in the JJ, high frequency thermal photon noise might limit our coherence times.

Referee #1

8. The conclusion mentions suppressed charge matrix elements as a key result, but this point is only addressed in the supplementary material. Given its importance, incorporating a brief discussion in the main text would strengthen the narrative and highlight its relevance to qubit coherence. It would also be useful to plot the expected T1 times vs flux for the quality factor quoted in the text in the same plot as their measured T1 times.

Our response:

We thank the referee for this constructive suggestion. We agree that the suppression of charge matrix elements is a central result of our work with direct implications for qubit relaxation. Specifically, we find that, in our model, the charged matrix elements are strongly reduced near the half-flux sweet spot. This suppression diminishes the dipole coupling of the qubit to charge noise and dielectric loss channels, thereby enhancing the relaxation time.

To explicitly demonstrate this, we compare our measured T_1 values with the expected T_1 values from a dielectric loss dominated model with

$$1/T_1 \propto E_C |n_{01}|^2 \tan\delta$$

where E_C is the charging energy, n_{01} is the charge matrix element and $\tan\delta = 1/Q$ is the dielectric loss tangent experimentally extracted in supplementary figure 7b. The measured and expected T_1 times coincide around the half-flux sweet spot, as shown below as a function of qubit frequency (also included in supplementary figure 7 in the revised manuscript). This good agreement with the dielectric loss model supports the suppression

of charge matrix element at half flux and identifies dielectric loss as the dominant relaxation mechanism at $0.5\Phi_0$.

Away from the sweet spot, the increase in charge coupling results in lower measured T_1 values than the dielectric-loss model values, suggesting that additional relaxation channels start contributing. In summary, we demonstrate that suppressed charge matrix elements play a protective role in qubit relaxation and presents an effective design principle for engineering qubits with longer lifetimes.

Referee #2 (Remarks to the Author):

The manuscript reports on a implementation of gatemon qubits using split-junction devices based on InAs/Al 2D heterostructures. The key advance is the demonstration of significantly enhanced anharmonicity (>100%) at the half-integer flux sweet spot, enabling fast and potentially more robust gate operations without requiring gate voltages. The authors also use high-resolution spectroscopy to extract detailed features of the current-phase relation, offering insight beyond standard transport techniques.

The results are timely and relevant to the community working on hybrid superconducting-semiconducting qubits and may present a significant step toward leveraging transparent junctions in scalable quantum hardware.

Our response:

We thank the referee for the positive assessment of the novelty and relevance of our results.

Referee #2

However, there are several points that should be addressed before the manuscript can be considered for publication.

Comments:

- The phrase "anharmonicity in excess of 100%" in the abstract is confusing. Anharmonicity is defined formally in Eq. (4), but the term is used repeatedly before that point in the manuscript. The authors should clarify this early in the text, preferably when the concept is first introduced. For instance, is 100% referring to the qubit frequency itself, or relative to the fundamental harmonic level spacing?

Our response:

We thank the referee for pointing this out. In the revised manuscript, we have introduced the definition of anharmonicity at the beginning of the manuscript (marked in red), where the concept is first introduced. In particular, we now explicitly state that an anharmonicity of 100% is defined relative to the qubit transition frequency f_{01} . For comparison, the qubit frequency in a standard AlOx transmon is given by $f_{01} \approx \sqrt{(8E_J E_C)/h} - E_C/h$, which typically amounts to an anharmonicity of only a few percent of f_{01} .

Referee #2

- In Fig. 1(d), the mapping between plotted lines and the left/right y-axes is unclear. The caption and figure labeling should be revised to indicate clearly which data correspond to which axis. As it stands, this figure is difficult to interpret.

Our response:

We thank the referee for the suggestion to improve figure 1d. In the revised figure, we have made the correspondence between the plotted data and the left/right y-axis clearer. In particular, we have added a new legend and improved axis color-coding to unambiguously indicate which data belongs to a particular axis. The key device parameters, i.e. anharmonicity and charge dispersion linewidths which are suppressed below 1 Hz, are now directly reported in the figure caption for better comprehensibility.

Referee #2

- The transition from Eq. (2) to Eq. (3) introduces the concept of multiple channels in the Josephson junction, but the text does not explain this development adequately. Please expand this section to clarify the physical picture, assumptions, and implications of including multiple conduction channels.

Our response:

We thank the referee for pointing out the need for more explanation between Eq. (2) and Eq. (3). In the revised manuscript, we have included a sentence to clarify the inclusion of multiple conduction channels in each JJ. Our junctions have a finite lateral width of ~ 800 nm, which allows for multiple transverse modes/channels to contribute to the supercurrent

across each JJ based on the lateral confinement. Each channel corresponds to an Andreev bound state (ABS) characterized by its transparency T_i . The total supercurrent is then the sum of the contribution from all these channels.

However, only a subset of these channels is highly transparent and effectively contributing to the supercurrent, thereby dominating Josephson transport across the junction (consistent with prior works in Mayer *et al.*, Nat. Comm. 11, 212 (2020); Goffman *et al.*, New J. Phys. 19, 092002 (2017)). In our single-transparency model, we categorize these dominant channels into an effective single-transparency channel with N representing the corresponding number of such highly transparent modes (see response for point 5 of Referee #1 on page 3 for further details). This assumption captures the essential nuances of multichannel transport dominated by the highly transparent modes, while keeping the model simple and easily applicable to experimental data.

Referee #2

- The coupling strength g is introduced in connection with the spectroscopy data, but its precise definition is missing. This is crucial, as multiple conventions exist in the literature. For example, whether the Hamiltonian term is written as $\hbar g(a^\dagger \sigma_- + a \sigma_+)$ or with $\hbar g/2$, or hg etc which all leads to numerical differences. Please state the exact convention used.

Our response:

We thank the referee for highlighting the need for precise definition of the coupling strength g . In our work, we use the Jaynes-Cummings Hamiltonian denoted by:

$$H = \hbar\omega_r a^\dagger a + \frac{\hbar\omega_{ge}}{2} \sigma_z + \hbar g(a^\dagger \sigma_- + a \sigma_+)$$

where a^\dagger, a are the photon creation and annihilation operators, σ_z, σ_\pm are the Pauli and ladder operators for the qubit, g is the vacuum coupling strength and ω_r, ω_{ge} are the resonator and the qubit transition frequency respectively. The resultant shape of the avoided crossing is then given by:

$$E_{\pm, n} = \hbar n \omega_r \pm \frac{\hbar}{2} \sqrt{(4ng^2 + \Delta^2)}$$

where $\Delta = \omega_r - \omega_{ge}$ denotes the detuning of the qubit from the resonator frequency.

Referee #2

- Furthermore, how is g extracted? Is it taken directly from the vacuum Rabi splitting at resonance, or obtained via a full fit of the qubit-resonator avoided crossing as a function of flux? For devices with broad linewidths, a full fit generally provides a more reliable estimate, and this should be clarified.

Our response:

In our devices, g is extracted by performing a full fit of the above equation for the avoided crossing spectrum shape to the experimental data. We now show the fits for our 3 devices

in supplementary figure 8. From the fits, we obtain $g = 122$ MHz for device 1, $g = 101$ MHz for device 2 and $g = 170$ MHz for device 3, respectively.

Referee #2

- The extracted values of g differ significantly between the three devices, despite nominally identical qubit designs and similar resonator frequencies. The authors should comment on this variation. Is it due to fabrication variation, junction asymmetry, or some other source? If it is due to the different matrix elements from the various junction transparencies, then it should be mentioned clearly.

Our response:

We thank the referee for highlighting the variations in the coupling strength g across devices. In general, the Josephson energy E_J depends on the JJ critical current I_c ($E_J = \hbar I_c / 2e$), which is further determined by the junction dimensions and the effective channel transparencies. Although the three devices are nominally identical in geometry and resonator frequency, mesoscopic variations in channel transparency due to microscopic variations in barrier thickness or disorder causes a substantial variation in I_c among the different devices. This changes the E_J/E_C ratio which directly affects the charge matrix element n_{01} . Since the coupling strength is directly proportional to the charge matrix element, i.e. $g \propto n_{01}$, any variation in the channel transparency translates into an observed change in g .

To explicitly quantify our claim, we extract the critical current I_c of the SQUID loop for device 1 and device 2 using the fit parameters in Table 1 of the main text. We obtain $I_{c1} \sim 147$ nA for device 1 and $I_{c2} \sim 65$ nA for device 2, resulting in $I_{c1}/I_{c2} \approx 2.26$. In the transmon regime with $E_J/E_C \gg 1$, the coupling strength g is related to E_J/E_C [Koch *et al.*, PRA 76, 042319 (2007)] by:

$$g \propto n_{01} \approx \left(\frac{E_J}{32E_C} \right)^{1/4}$$

which results in:

$$\frac{g_1}{g_2} = \left(\frac{I_{c1}}{I_{c2}} \right)^{1/4}$$

For $I_{c1}/I_{c2} \approx 2.26$ in our work, we obtain $g_1/g_2 = 1.21$, which is in agreement with the measured coupling strengths of $g_1 = 122$ MHz for device 1 and $g_2 = 101$ MHz for device 2, ultimately validating our argument that different matrix elements due to variations in the JJ transparencies predominantly lead to the observed spread in g .

Referee #2

- There appears to be a missing T_2 in the inline equation before Table 1, which makes the expression incomplete. Please correct this.

Our response:

We thank the referee for pointing this out. We have included the missing T_2 in the revised manuscript.

Referee #2

- The manuscript claims suppression of the charge matrix element near the sweet spot. However, this does not seem to be supported by experimental evidence in the current work. In the supplemental material, a theoretical calculation is shown, but the flatness of T_1 -times across the sweet spot implies no observable suppression. Do the authors see any change in Rabi frequency that might indicate such suppression? If so, they should quantify this, but note that such a measurement would require a careful calibration of the entire drive line including PCB and device response. Without this, the claim should be moderated.

Our response:

We thank the referee for this very insightful comment. In our dielectric-loss dominated model, the T_1 relaxation time is given by:

$$1/T_1 \propto E_C |n_{01}|^2 \tan\delta$$

where E_C is the charging energy, n_{01} is the charge matrix element and $\tan\delta = 1/Q$ is the dielectric loss tangent. Typically, $\tan\delta$ does not have a strong frequency/flux dependence, therefore any variations of T_1 with flux is primarily determined by the charge matrix element n_{01} .

To verify this relation, we compare the model-predicted charge matrix elements with values extracted from our measured T_1 values. To eliminate the unknown prefactors, including the assumed constant loss tangent, we normalize both the theoretical and experimental values to their respective values at half-flux sweet spot. The resulting plot for $n/n_{01,HFQ}$ vs flux is shown below:

The model predicts a suppression of n_{01} at $\Phi = 0.5\Phi_0$, corresponding to an enhancement in T_1 , and an increase in n_{01} away from the sweet spot. In contrast, our measured T_1 values

exhibit significant fluctuations, which obscures this dependence. Nevertheless, the predicted variation of n_{01} lies within the envelope of the experimental fluctuations, signifying that the experimental data is consistent with – but do not directly resolve – the expected suppression.

In the revised manuscript, we now explicitly mention that the suppression of the charge matrix elements could not be experimentally resolved in our present data. As suggested by the referee, a fully calibrated Rabi frequency measurement, which accounts for entire the fridge drive lines and packaging, would be required for a quantitative validation.

Referee #2

The work presents interesting and potentially impactful experimental results that merit publication in Nature Communications after the above issues have been addressed. I recommend revision to clarify the points above.

Our response:

We thank the referee for their time and careful review of our work and finding our experimental results interesting and impactful.

Reviewer #3 (Remarks to the Author):

Our response:

We thank the referee and their co-reviewer for their participation and time in carefully peer reviewing our manuscript.

Referee #4 (Remarks to the Author):

The authors of the manuscript “Liu et al., Strongly-anharmonic gateless gatemon qubits based on InAs/Al 2D heterostructure,” study the properties of transmon devices in which the qubit Josephson junction has been replaced by an asymmetric DC SQUID with semiconductor Josephson junctions (Sm-JJs). The Sm-JJs are characterized by a current-phase relation (CPR) that contains higher harmonics in phase, ϕ . When the SQUID flux is fixed at half of a flux quantum ($\Phi^0/2$), the supercurrent interference partially cancels the $\cos \phi$ -component of the SQUID energy-phase relation. This results in a modified transmon spectrum with increased anharmonicity while, at the same time, retaining an acceptable charge dispersion linewidth. The authors report measurements of the qubit dispersion as a function of SQUID flux on three different devices. Next, they fit the transmon spectra, replacing the Josephson energy term by an ansatz for the flux/phase-dependent energy of a SQUID with two junctions. They compare two different approaches. The first one uses a

parametrization of the junction energy-phase relation that assumes N_i identical channels of transparency T_i and gap Δ in the short channel limit (following Beenakker, Ref 37). The other model constitutes of a series of $\cos(n\phi)$ -harmonics up to eighth order. Both models yield fits of roughly similar accuracy. The authors proceed to plot the SQUID CPR and the CPRs of the individual junctions that correspond to the extracted model parameter. Finally, they report the results of the qubit characterization for SQUID flux $\Phi_0/2$, extracting the characteristic decoherence times T_1 and T_2^* .

Our response:

We thank referee for precisely summarizing our work.

Referee #4

The work addresses one of the current big topics in the field of solid-state quantum information processing: the hunt for qubit systems with long coherence times and short gate operation times. The superconducting transmon qubit has the advantage of reduced charge sensitivity, but its small spectral anharmonicity increases the time-scale of gate manipulations. In this work, the authors make use of higher harmonics in the potential of a SQUID with Sm-JJs to engineer a qubit spectrum with larger anharmonicity. Whereas the idea of using higher harmonics is not entirely new, the authors present a neat demonstration of the concept that will be of interest to the wider community.

Our response:

We thank the referee for the positive assessment on the quality of our results.

Referee #4

I read the manuscript with great interest but think that the presentation and analysis lack depth. Additionally, important information is missing, which precludes the reader from reaching their own conclusions. In its current state, I cannot recommend the paper for publication in Nature Communications.

Our response:

We are very grateful to the referee for their time to review our work. We note that the listed points are important technical issues, the corrections of which have significantly improved our manuscript. We hope the new heavily presented information will convince the referee to recommend publication.

Referee #4

Let me elaborate:

The data in Tab. 1 are not discussed.

The choice of parametrization of the junction potential $U(\phi, T)$, implies that the supercurrent transport can be modeled as identical channels in the short junction limit. Taking a look at the results of the fit (Tab. 1), I am wondering if this makes sense. I am particularly intrigued by the transparency values, $T \sim 0.928-0.999$. These are incredibly large values for semiconductor Josephson junctions, for which one expects a sizable mismatch between the

Fermi surfaces of semiconductor and metal, consequently interface scattering and lower transparencies.

- Have the authors critically assessed the interpretation of highly transmitting channels?
- Has the interface quality been studied in similar JJs? (References?)

Our response:

We thank the referee for raising this important point. The transparencies extracted in table 1 are indeed very high ($T \sim 0.93 - 0.999$), and we have carefully examined their interpretation. We emphasize that these values do not represent the literal transmission probabilities of every microscopic channel, but rather as effective parameters within our short-junction, single-transparency model. More importantly, when we allow for a distribution of channel transparencies in the fit, the population consistently converges into a single dominant, high-transparency value (please see response on point 5 of Referee #1 for additional details), demonstrating that the qubit spectra are well captured by few channels with an average effective transparency close to unity.

Experimentally, the qubit spectra reveal a strongly forward-skewed CPR. Such strong forward skewness arises only when higher harmonics are prominent in the CPR, possible only at high transparencies - signatures of highly transmitting channels in the short-junction limit. This close agreement between the observed CPR shape and our short-junction model provides robust independent evidence of near-unity channel transparencies in our devices.

Physically, such high effective transparencies are consistent with the materials system we employ in our work. InAs-Al 2DEGs grown by in-situ molecular beam epitaxy (MBE) exhibit atomically abrupt, epitaxial interfaces with strong hybridization between the semiconductor and the Al layer. This minimizes Fermi-surface mismatch and interface scattering, enabling exceptionally transparent supercurrent transport. Prior works on similar epitaxial junctions have shown skewed CPRs and near-unity transmissions [Mayer *et al.*, Nat. Comm. 11, 212 (2020); Goffman *et al.*, New J. Phys. 19, 092002 (2017); Kjaergaard *et al.*, Nat. Comm. 7, 12841 (2016)]. In addition, tunneling electron microscopy (TEM) and spectroscopy studies have further confirmed the atomic-scale uniformity due to lattice alignment and absence of any interfacial oxide in such heterostructures [Shabani *et al.*, PRB 93, 155402 (2016)], providing direct evidence of the high interface quality needed for such near-unity transparency.

In summary, although microscopic variations in individual channel transmissions exists, the combination of skewed CPR in our measured data, fits that consistently converges to a single dominant transparency and prior material platform studies support the conclusion that supercurrent transport in these devices is dominated by a few nearly ideal channels.

Referee #4

- Are the Sm-JJs indeed in the short junction limit? I.e., $L \ll \xi \approx \hbar v_F / 2\Delta$? ($L \equiv$ electrode separation; $v_F \equiv$ Fermi velocity in the semiconductor)

Our response:

We thank the referee for this important question, as the validity of the short-junction approximation is indeed central to our modeling of the CPR. The junction length in our devices is $L = 250$ nm. Using the extracted electron carrier density $n_s \approx 7.68 \times 10^{11} \text{ cm}^{-2}$ of the InAs 2DEG, determined during the heterostructure growth, we obtain the Fermi velocity as $v_F \approx 1.17 \times 10^6 \text{ m/s}$. With the induced Al superconducting gap ($\Delta \approx 230 \text{ } \mu\text{eV}$), we obtain a coherence length of:

$$\xi = \frac{\hbar v_F}{\pi \Delta} \sim 1 \text{ } \mu\text{m}$$

which gives $L/\xi \sim 0.25$, placing our devices in the crossover of the short and intermediate regime.

This is further supported by the observation of strongly forward-skewed CPR, indicative of high transparency channels with substantial higher-harmonic content – clear hallmarks of the short-junction limit (please see response to point 4 of Reviewer #1 for additional details). In addition, both an empirical harmonic expansion that allows us to extract a model-independent CPR alongside our short-junction single transparency model show excellent agreement with the experimental CPR, confirming that the short-junction approximation justifiably and sufficiently captures the dominant physics of our junctions.

Referee #4

I didn't find the value of Δ in the text. Has it been measured or inferred from the critical temperature of the Al layer?

Our response:

We thank the referee for pointing this out. The induced superconducting gap of the Al layer is $\Delta \approx 230 \text{ } \mu\text{eV}$, determined from its critical temperature $T_c \sim 1.5\text{K}$. This value was measured on a similar InAs-Al heterostructure grown in the same MBE chamber using an identical in-situ process [Yuan *et al.*, J. Vac. Sci. Technol. A 39, 033407 (2021)]. We note here that in our model, the Δ is absorbed into the product $N \times \Delta$ as one single fit parameter in table 1, where N is the effective number of channels.

Referee #4

- Do the values N match expectations based on lateral size quantization?

Our response:

We thank the referee for this insightful question. For a junction width of $W = 800$ nm and measured InAs 2DEG density of $n_s \approx 7.68 \times 10^{11} \text{ cm}^{-2}$, the number of propagating transverse modes based on geometric lateral confinement is given by:

$$N \approx \frac{2k_F W}{\pi}, \quad k_F = \sqrt{(2\pi n_s)}$$

where the factor of 2 arises due to spin degeneracy and k_F is the 2D Fermi wavevector. This yields $N \sim 110$ available modes, consistent with expectations for InAs-based heterostructures. More importantly, this value represents the maximum number of available

transverse modes. In practice, only a small subset of these channels are strongly coupled to the superconductor, thereby making them highly transparent and contributing significantly to the Josephson current. Various factors such as disorder, electrostatic edge depletion while wet-etching, nonuniform lead coupling, strong mode-dependent transparencies etc. typically reduce the effective number of contributing channels.

Several prior works have emphasized this distinction. Mayer *et al.* [Nat. Commun. **11**, 212 (2020)] and Costa *et al.* [Phys. Rev. B **108**, 054522 (2023)] showed that the current–phase relation in InAs–Al JJs is governed primarily by a small number of high-transmission modes, despite the presence of many geometric channels. Similarly, Goffman *et al.* [New J. Phys. **19**, 092002 (2017)] directly measured the channel distribution in InAs–Al nanowire JJs, finding that only a few channels dominate the supercurrent. Earlier theoretical work by Volkov *et al.* [Physica C: Superconductivity **210** (1993)] also predicted that proximity coupling to superconductors selectively enhances the contribution of certain modes.

In conclusion, our fitted values of $N \sim 10 - 12$ corresponds to the effective number of highly transmissive Andreev channels, consistent with the quasi-ballistic transport regime of our devices. This interpretation is in agreement with theoretical expectations and previous experimental reports, providing a coherent and physically grounded description of our JJs.

Referee #4

- It is mentioned that all Sm-JJs are nominally identical. Why is there are large spread in the parameter values of Tab. 1? What are the values of the critical current for each junction?

Our response:

We thank the referee for highlighting this variation in the parameter values of table 1. Although the three devices are nominally identical in geometry and fabrication, mesoscopic variations in channel transparency due to device-to-device microscopic variations in material properties such as barrier thickness or disorder causes a substantial spread in the extracted parameters. Therefore, the reported values in table 1 represents junction-specific fit parameters within the same device, capturing this natural variation. From our fits, the critical current for each JJ is: $I_{c1} \approx 178.9 \pm 19.7$ nA for JJ1 and $I_{c2} \approx 71.9 \pm 1.1$ nA for JJ2 in device 1, and $I_{c1} \approx 138 \pm 17.6$ nA for JJ1 and $I_{c2} \approx 100.8 \pm 1.2$ nA for JJ2 for device 2. For the SQUID loop, we obtain $I_{c1} \sim 147.2 \pm 23.4$ nA for device 1 and $I_{c2} \sim 65.3 \pm 14.8$ nA for device 2, respectively.

These extracted values from our fits can be independently verified with the measured variations in the qubit-resonator coupling strength g , which depends on the Josephson energy E_J and, in turn, the critical current I_c via the relation: $g \propto I_c^{1/4}$ [Koch *et al.*, PRA **76**, 042319 (2007)] (see response to Referee #2 on page 10 for details). For $g_1/g_2 = 1.21$ measured in our devices, as shown in supplementary figure 8, we obtain $I_{c1}/I_{c2} \approx 2.26$, consistent with the extracted values of I_c for the SQUID loops in device 1 and 2. Thus, the spread in I_c and the corresponding fit parameters are real and physical, and not an error.

Referee #4

- What is the mean free path in the semiconductor material? Can the authors rule out the diffusive transport limit?

Our response:

We thank the referee for this important question. The mean free path of the InAs 2DEG is given by:

$$l = \frac{\hbar k_F \mu}{e}$$

where $k_F = \sqrt{2\pi n_s}$ is the 2D Fermi wavevector, n_s is the electron carrier density and μ is the electron mobility. Using the measured values of $n_s \approx 7.68 \times 10^{11} \text{ cm}^{-2}$ and $\mu \approx 20,000 \text{ cm}^2/\text{Vs}$, we obtain $l \approx 290 \text{ nm}$. Since the junction length is $L = 250 \text{ nm}$, the ratio $l/L \approx 1.2$ places our device in the quasi-ballistic regime, but well away from the strongly diffusive limit ($l \ll L$). In this regime, many electron trajectories can traverse the junction without any scattering, giving rise to ballistic contributions to transport, while a non-negligible fraction will scatter, resulting in only a few effective high-transparency channels that dominate supercurrent transport. Furthermore, the Al coherence length is $\xi \sim 1 \mu\text{m}$, ensuring that the device also lies in the short-junction regime. In conclusion, although diffusive contributions to transport cannot be entirely excluded, the combination of length scales indicates that our junctions operate in the quasi-ballistic regime, consistent with transport picture presented in the manuscript. A more quantitative determination could be obtained by directly comparing the normal-state resistance to the Sharvin resistance, which we view as a natural extension beyond the scope of the present qubit spectroscopy devices.

Referee #4

I recommend the authors to provide this information in the revised manuscript and include references to available literature.

Our response:

We thank the referee for the suggestion to include more information. We have added the above points along with the necessary references in the revised manuscript.

Referee #4

There are several aspects of the CPR discussion (page 12f and Fig.4) that I find strange: In Fig. 4, the normalized CPRs of JJ2 of Device I and II match. They are calculated based on the fit parameters in Table 1. However, judging by the transparency values given there, I would expect JJ1 of Device II to match with JJ2 of Device I. Are the labels in Fig. 4 wrong, or am I missing something in the analysis? Is there a way to know which one of the two physical junctions has the parameters of JJ1 and JJ2 in this experiment? Or are the labels interchangeable?

Our response:

We thank the referee for raising this question. In our analysis, the labels JJ1 and JJ2 are assigned by comparing the extracted critical currents of the two junctions obtained from the CPR fits. Specifically, we define the junction with the larger critical current as JJ1. Beyond this convention, the junction properties are interchangeable, i.e. swapping the labels does not alter the simulated qubit spectrum or the interpretation of the results. Since the two junctions are fabricated to be nominally identical, their distinction arises only from the physical parameters extracted through our modeling. Therefore, the notation JJ1 and JJ2 can be used interchangeably without affecting any conclusions of our work.

Referee #4

What do we learn from Fig. 4c? The authors provide a plot of the first 16 amplitudes of the Fourier series expansion of eq. 2 for the parameters T_i in Tab. 1. In other words, an (effectively) one-parameter expression (as N_i and Δ drop out by normalization) is recast into another expression with 16 parameters. It is stated that the magnitudes of the amplitudes strongly decrease with increasing index n . This is true but easily understood when we look at the Fourier series of a sawtooth waveform, $|A_n| \propto 1/n$, which represents the extreme limit of a skewed CPR. I don't see the value of this analysis.

Our response:

We thank the referee for this valuable comment. Indeed, the overall transport regime can already be inferred from the skewness of the CPR: a sinusoidal CPR reflects diffusive transport, while forward skewness signals the presence of high-transparency modes. The value of the harmonic expansion, however, lies in going one step further - the relative ratios of the harmonic terms A_n/A_1 directly connect the microscopic transport properties of the JJ to the shape of the Josephson potential landscape. This information is crucial for qubit engineering, since the higher harmonics reshape the potential beyond the simple cosine form and thereby determine the anharmonicity, charge dispersion and flux-noise sensitivity of the qubit.

In our devices, we find $A_2/A_1 \approx 1/2$ for the SQUID loop and $A_2/A_1 \approx 1/6$ for the individual junctions, together with substantial higher harmonics. These values indicate that the Josephson potential substantially deviates from the standard $\cos(\phi)$ form and acquires additional $\cos(2\phi)$ and higher-order contributions. As shown in figure 1c, the Josephson potential now approaches a π -periodic landscape, which makes the curvature of the potential minima steeper and more asymmetric within each 2π phase period. This modified curvature naturally reduces the qubit's charge dispersion, since the multi-well potential suppresses the sensitivity to offset charge due to suppressed charge matrix elements, and explains the strongly anharmonic spectra observed in our devices.

More broadly, tuning the harmonic ratios A_n/A_1 provides a versatile design tool for engineering novel qubit designs. For example, complete suppression of the first harmonic, i.e. $A_1 = 0$, yields a $\cos(2\phi)$ potential with exact π -periodicity, enabling a $\cos(2\phi)$ qubit with degenerate ground states and enhanced T_1 relaxation time at the half-flux sweet spot [Larsen et al., PRL 125, 056801 (2020)]. Such qubits can be realized by naturally extending our design

to symmetric SQUIDs where the first harmonics of the two JJs cancel each other. Similarly, devices with $A_1 \ll A_2$ and $A_2 < 0$ satisfy the conditions for novel “harmonium” qubits, which feature non-degenerate but noise-resilient ground states [Hays et al., arXiv:2502.15459 (2025)]. Our devices, with intermediate ratios such as $A_2/A_1 \approx 1/2$, showing partial suppression of the charge matrix elements at half-flux sweet spot, naturally fall within this broader design landscape, and illustrate how harmonic content extracted from transport directly connects to qubit performance and engineering of future qubit modalities.

Referee #4

On the other hand, in Fig. 3, the authors compare the single-transparency fit with a model for USQUID composed of eight $\cos(\phi)$ -harmonics. How do the coefficients of the Fourier series expansion (A_n) in Fig. 4c compare to the eight fitting parameters of the second model? Can we learn something about the junction physics or the fitting procedure by comparing the numbers?

Our response:

We thank the referee for this insightful question. The comparison between the Fourier coefficients in figure 4c and the eight-harmonic model (4 harmonics per JJ) for each individual JJ in device 1 and 2 is shown below:

However, we note that the two representations are not directly comparable term by term, as they are obtained using different approaches and assumptions: The Fourier coefficients result from a direct numerical decomposition of the extracted CPR, while the eight-harmonic model corresponds to a fit in which the amplitudes of the first four harmonics of each JJ are treated as free parameters. Nevertheless, comparing the two representations still provides valuable insight.

In particular, it shows that higher-order harmonics $A_{n>4}$, although much smaller in magnitude than the leading terms, still play a significant role in accurately describing the

qubit spectrum and the resulting fits. Their inclusion systematically reduces the fitting residuals and improves agreement with the measured transition frequencies. This is evident in figure 4d, 4e and 4f, where the residuals of the eight-harmonic model fluctuate around zero (oscillatory behavior) in contrast to the single transparency model fit. Therefore, higher harmonics cannot be neglected in the modeling. From a junction-physics perspective, their inclusion and identification reinforce the interpretation of highly transparent Andreev channels, rather than behaving like tunnel junctions.

Referee #4

I disagree with the statement “More notably, the differences in the CPR and harmonic contributions between individual JJs are minimal, enabling us to extract fine details in the current-phase relations, far surpassing the capabilities of any standard transport measurement.”

The first part is wrong. It is observed that the CPRs of two junctions are identical. The other two are substantially different. Therefore their harmonic content is different. The second part is not discussed in the manuscript, and I highly doubt it is true.

Our response:

We thank the reviewer for this helpful comment. We agree that our original statement was misleading. While two JJs exhibit nearly identical CPR, others show substantial differences, resulting in distinct harmonic content that reflects variations in microscopic junction parameters. We have revised the text to reflect this observation accurately, without implying uniformity across all measured junctions.

Regarding the claim about surpassing transport measurements, we do not intend to claim that our approach universally provides more information than all standard transport techniques. Rather, the harmonic decomposition of the CPR offers complementary insights into the microscopic transport regime, Josephson potential landscape and qubit properties, which ultimately affect the qubit’s performance, that are not directly accessible from DC I-V characteristics alone. In the revised manuscript, we now include: “*While two junctions exhibit similar CPRs, others show substantial deviations, leading to distinct harmonic content. This diversity reflects the sensitivity of the CPR to microscopic junction parameters. The harmonic analysis provides complementary information to standard transport measurements, allowing one to identify deviations from purely sinusoidal behavior and to engineer the Josephson potential landscape, thereby tailoring qubit properties for better performance.*”

Referee #4

Excluding other systematic errors for now, the accuracy of the CPR extraction depends on the choice of a suitable parametrization and the quality of the fit to the experimental data. The authors have omitted a good deal of error analysis here. What are the confidence intervals for the parameters in Tab. 1?

Our response:

We thank the referee for pointing out the need for confidence intervals and error analysis. The confidence intervals of the fitting parameters in table 1 are obtained from the covariance matrix of the nonlinear least-squares fit. We follow White’s heteroskedasticity-consistent method [H. White, *Econometrica*, vol. 48, no. 4, pp. 817–838 (1980)], where we first compute the Jacobian matrix J of the fit numerically using a finite-difference scheme. The covariance matrix C is then approximated as:

$$C = \epsilon^2 (J^T J)^{-1}$$

where ϵ is the mean standard error of the fit. The 95% confidence interval for each fitted parameter β_i is then given by:

$$\beta_i \pm Z \sqrt{C_{ii}}$$

where $Z \approx 1.96$ is the Z-score that corresponds to 95% confidence interval. The resulting fit parameters with their 95% confidence intervals is listed below:

Table of 95% confidence interval for Device I and II

Device	$N_1 \Delta$ (GHz)	T1	$N_2 \Delta$ (GHz)	T2	E_C (GHz)
I	221.0 ± 22.9	0.999 ± 0.099	98.9 ± 2.9	0.926 ± 0.006	0.197 ± 0.003
II	138.0 ± 3.0	0.928 ± 0.006	467.0 ± 88.5	0.542 ± 0.095	0.163 ± 0.002

We have now added the confidence intervals in table 1 of the revised manuscript.

To propagate these uncertainties to the extracted CPR and the corresponding Fourier harmonics, we employ a bootstrapping scheme with 500 Monte-Carlo samples. Each parameter β_i is assumed to follow a normal distribution with standard deviation $\sigma = \sqrt{C_{i,i}}$, which is justified by the large number of data points relative to the number of fit parameters. Unphysical parameters, such as transmission exceeding unity, are excluded by post-selection. For each valid sample, the CPR is then computed numerically on a uniform grid of 200 phase points, and the harmonic amplitudes are extracted using a Fast Fourier Transform (FFT). The reported error bars for the current and the harmonic amplitude, as shown in the responses below, correspond to the median 95% interval of the resulting distributions.

Referee #4

How large are the error bars in Fig. 4a,b?

Our response:

Using the procedure described above, we extracted the normalized CPR of the individual JJs in device 1 and the corresponding SQUID loop. The resulting plots, including error bars depicted by the shaded area, are shown below (also shown in supplementary figure 9 of the revised manuscript):

Similarly, for device 2, we obtain:

Referee #4
In Fig. 4c?

Our response:

Figure 4c with error bars included for the individual JJs and the SQUID loops are shown below:

Referee #4

Can the authors provide examples by detailed comparison with transport experiments in the literature to support their statement?

Our response:

The parameters extracted from our fits in table 1 yield transparencies and CPR skewness consistent with prior transport measurements. Specifically, we obtain a small number of highly transparent channels with transmission probabilities $T \geq 0.92$, leading to strongly forward-skewed CPRs with $A_2/A_1 \sim 0.2 - 0.5$. These values are in excellent agreement with previous transport studies in similar hybrid systems- for example, Kjaergaard *et al.* [Nat. Comm. 7, 12841 (2016)] reported forward-skewed CPRs in epitaxial InAs-Al JJs consistent with $T \geq 0.9$; Mayer *et al.* [Nat. Comm. 11, 212 (2020);] observed skewness corresponding to few-mode transport with $T \approx 0.6 - 0.9$; and Goffman *et al.* [New J. Phys. 19, 092002 (2017)] demonstrated distributions with near-unity transparency channels in atomic-scale contacts. Our results therefore lie within the established transport regime of highly transparent, quasi-ballistic junctions, while harmonic analysis provides a complementary and independent confirmation of these properties.

Referee #4

Finally, the authors should provide an analysis of systematic errors in their experiment. E.g., generally, the phase drop across the SQUID loop inductance must be taken into account when determining the CPR. If the inductance is large, the shape of the CPR has to be determined self-consistently. Have the authors calculated/estimated the loop inductance (geometric and kinetic components) and ascertained that it is safe to omit it from their analysis? How would it affect the precision of their CPR extraction and the fit values for transmission in Tab. 1?

Our response:

We thank the referee for highlighting the role of the loop inductance and its potential influence on the CPR extraction. We first carry out a quantitative estimate of the geometric and kinetic inductance of the SQUID loop and its associated energy scale, followed by an explicit evaluation of the screening parameter β .

Typically, the geometric loop inductance for a square loop with rectangular conductors is estimated by [Grover, Inductance Calculations: Working Formulas and Tables, Dover (1946)]

$$L_{geo} = \mu_0 a \left[\ln \left(\frac{2.64a}{\omega + t} \right) + 0.5 \right]$$

where a is the length of one side of the SQUID loop and ω, t are the width and thickness of the Al layer. For our geometry of $a = 50 \mu m$, $\omega = 8 \mu m$, $t = 30 nm$, we obtain $L_{geo} \sim 207 pH$. Kinetic inductance for such structures with 30 nm Al is typically around 5% of L_{geo} , resulting in a total inductance of $L \sim 220 pH$. The corresponding loop inductive energy $E_L = \Phi_0^2 / 2L \gg 1000 GHz$. For reference, the Josephson energy of the SQUID loops is $E_J \sim 75 GHz$ for device 1 and $E_J \sim 33 GHz$ for device 2, while the charging energy is $E_C \sim 200 MHz$. Therefore, the loop inductance provides an energy scale several orders of magnitude larger than both the Josephson and charging energies, thereby:

$$E_L \gg E_J \gg E_C$$

This firmly places our devices in the regime where the loop acts as a stiff phase bias, and inductive corrections to the CPR are expected to be very small.

In addition, a complementary way to assess the relevance of the loop inductance is to estimate the screening parameter

$$\beta = \frac{2\pi L I_c}{\Phi_0}$$

which results in an extra phase shift due to the current I_c flowing in the SQUID loop, i.e.

$$\Delta\phi = \frac{2\pi\Phi_{ext}}{\Phi_0} - \frac{2\pi L I_c}{\Phi_0}$$

Using L and I_c determined above, we obtain $\beta \approx 0.09$ for device 1 and $\beta \approx 0.04$ for device 2. Both the values are well below unity, meaning the system operates deeply in the weak-screening regime. In this limit, the inductive term enters only as a small phase shift, i.e. $\Delta\phi \sim \beta$, and produces very minimal corrections to the CPR harmonics.

In summary, the inductive energy is far larger than both E_J and E_C . The associated screening parameter $\beta \ll 1$, thereby ascertaining that the precision of our CPR extraction is robust.

Referee #4

Additional comments:

There is a reference missing on page 16, bottom line.

Our response:

We thank the referee for pointing this out. We have added the missing reference in the revised manuscript.

Referee #4

I, personally, find expressions like “gateless” gatemon or “super-semi” Josephson junctions confusing and distracting. The oxymoron in the title made me skip the paper when it first appeared on the arXiv. I presume other researchers feel similarly about colloquialisms.

Our response:

We thank the referee for this comment. We have changed the term “gateless gatemon” to “flux-tunable transmon” and “super-semi” to “superconducting-semiconducting” in the revised manuscript.

Referee #4

The authors state that their transmon qubit allows for faster gate operations than traditional transmon and gatemons. As someone who works in the periphery of the quantum information field, I would greatly appreciate if the authors gave more context and examples. It would benefit the general readership.

Our response:

We thank the referee for the valuable suggestion to clarify our statement. We note that conventional AlOx-based transmons can achieve sub-10 ns gates using advanced pulse shaping protocols such as DRAG [Motzoi *et al.*, PRL 103, 110501 (2009); Lucero *et al.*, Phys. Rev. A 82, 042339 (2010)]. Our point here is not to emphasize that such gate times are unattainable in other platforms, rather that our device reaches comparable speeds intrinsically, without relying on advanced DRAG protocols.

The key advantage of our architecture is the strongly anharmonic qubit spectra engineered by the higher-harmonic contributions. This produces an energy level spectrum where the f_{01} transition is well separated from higher levels, intrinsically suppressing leakage under strong drives. As a result, we can apply large drive amplitudes and achieve raw Rabi frequencies exceeding 100 MHz using simple Gaussian-like pulses, while maintaining low leakage. In contrast, standard transmons with ~ 250 MHz anharmonicity experience significant leakage at these drive rates, and therefore require careful DRAG calibration to compensate.

Thus, the contribution of our work is not that we surpass the absolute gate speed demonstrated in optimized AlOx devices, but that we offer a simplified and robust route to fast gates. By embedding large anharmonicity directly into the device, our qubit design reduces the need for complex pulse shaping, minimizes calibration overhead, and naturally supports fast, high-fidelity gate operations.

Reviewer #1 (Remarks to the Author):

The authors have adequately addressed the concerns and comments from the previous round of review.

Our response:

We thank the referee for the positive assessment of the previous round of review.

Reviewer #1 (Remarks to the Author):

I do have one significant issue with their new revision regarding the suppression of the charge matrix element from their dielectric loss model. Their procedure seems to be circular - they extract the dependence of the loss tangent at each frequency using their measured T1 and then claim good agreement between their model, which uses the extracted loss tangent, and their experimental T1. The typical method for comparing with a loss model is to consider the expected functional form of T1 for some loss model, which includes the matrix element and a well-motivated frequency-dependent form for the loss tangent, and check whether the data can be fit using that model. The authors need to either remove the claim that their model fits their data well or provide reasoning for why their procedure is physically sound.

Our response:

We thank the referee for highlighting the circular nature in our earlier approach for determining the charge matrix element from the dielectric loss model. In the revised manuscript, we now adopt a weakly frequency dependent dielectric loss model based on previously reported measurements. We assume that the dielectric loss tangent follows the empirical relation:

$$\frac{\tan \delta (\omega)}{\tan \delta (\omega_0 = 2\pi \times 2.63 \text{ GHz})} = \left(\frac{\omega}{\omega_0} \right)^\varepsilon$$

with $\varepsilon = 0.15$ as reported by Nyugen *et. al.* [Phys. Rev. X 9, 041041 (2019)]. The reference loss tangent value of $\tan \delta (\omega_0 = 2\pi \times 2.63 \text{ GHz}) = 2.44 \times 10^{-4}$ corresponds to typical dielectric losses for InP substrates [Strickland *et. al.*, Phys. Rev. Res. 6, 023094 (2024)].

At the finite operating temperature of our qubit system $T = 50 \text{ mK}$, the dielectric loss limited relaxation time is given by:

$$T_{1,\text{diel}}(\omega) = \frac{1}{16\pi E_c \tan \delta (\omega)} \cdot \frac{1}{n_{01}^2} [1 + \coth(\hbar\omega/2k_B T)]^{-1}$$

The resulting plot for the reference quality factor $Q_{\text{ref}} = 1/\tan \delta (\omega_0 = 2\pi \times 2.63 \text{ GHz}) = 4100$ is plotted in the revised figure below (also included in supplementary figure 7c).

Although our measured T₁ times show no clear frequency dependence, indicating that the measured data is not sufficiently constraining, the results are consistent with a dielectric loss model at finite temperature of $T = 50 \text{ mK}$. This demonstrates that our analysis does not rely on circular fitting and that the observed relaxation times are physically consistent with realistic dielectric losses.

Reviewer #2 (Remarks to the Author):

The authors have addressed the previously raised points from myself and the other referees and I recommend publication.

Our response:

We thank the referee for their time and careful review of our manuscript and the recommendation for publication in Nature Communications.

Reviewer #3 (Remarks to the Author):

Our response:

We thank the referee and their co-reviewer for their participation and time in carefully peer reviewing our manuscript.

Reviewer #4 (Remarks to the Author):

The authors have addressed all points that needed clarification in their rebuttal letter.

Our response:

We thank the referee for the positive assessment of the rebuttal letter.

Reviewer #4 (Remarks to the Author):

I noticed, however, the sentence "This approach is justified for atomically clean JJs, where channel-to-channel variations are minimal." has been added on page 11 of the revised manuscript. I think this statement is misleading. Firstly, in their rebuttal letter, the authors point out that the variations in the critical current of the individual junctions likely originate from disorder or some other mesoscopic imperfections. Secondly, only about 1/10 of ~110 modes are highly transmissive due to either selective coupling or some sort of disorder/mesoscopic imperfection. I recommend this information be stated clearly (including the references on selective coupling and mode transmissions provided in the rebuttal letter) for the benefit of the reader.

Our response:

We thank the referee for this insightful comment. In the revised manuscript, we have now clarified that only a small fraction of the total transmitted channels, i.e. approximately 10%, are highly transmissive, while the majority exhibit reduced transparency due to mesoscopic imperfections such as disorder and selective mode coupling at the interface. We have also included the relevant references [34] and [39-41] in the revised manuscript to support this discussion.

Therefore, the revised text now states: *"This approach is justified for atomically clean JJs, where channel-to-channel variations are expected to be minimal. However, in practice, only a fraction of the total channels (~10% in our devices) is highly transmissive due to mesoscopic imperfections such as disorder and selective mode coupling."*

Reviewer #4 (Remarks to the Author):

I support the publication of the paper in Nature Communications.

Our response:

We thank the referee for the recommendation to publish the manuscript in Nature Communications.

Reviewer #1 (Remarks to the Author):

The authors have addressed my comments in the revised version of their manuscript.

Our response:

We thank the referee for their time and careful review of our manuscript.

Reviewer #3 (Remarks to the Author):

Our response:

We thank the referee and their co-reviewer for their participation and time in carefully peer reviewing our manuscript.